# Fluctuating climate and dietary innovation drove ratcheted evolution of proboscidean dental traits

**Juha Saarinen** [1] & **Adrian M. Lister** [2]

Identification of the selective forces that shaped adaptive phenotypes generally relies on current habitat and function, but these may differ from the context in which adaptations arose. Moreover, the fixation of adaptive change in a fluctuating environment and the mechanisms of long-term trends are still poorly understood, as is the role of behaviour in triggering these processes. Time series of fossils can provide evidence on these questions, but examples of individual lineages with adequate fossil and proxy data over extended periods are rare. Here, we present new data on proboscidean dental evolution in East Africa over the past 26 million years, tracking temporal patterns of morphological change in relation to proxy evidence of diet, vegetation and climate (aridity). We show that behavioural experimentation in diet is correlated with environmental context, and that major adaptive change in dental traits followed the changes in diet and environment but only after acquisition of functional innovations in the masticatory system. We partition traits by selective agent, showing that the acquisition of high, multiridged molars was primarily a response to an increase in open, arid environments with high dust accumulation, whereas enamel folding was more associated with the amount of grass in the diet. We further show that long-term trends in these features proceeded in a ratchet-like mode, alternating between directional change at times of high selective pressure and stasis when the selective regime reversed. This provides an explanation for morphology adapted to more extreme conditions than current usage (Liem's Paradox). Our study illustrates how, in fossil series with adequate stratigraphic control and proxy data, environmental and behavioural factors can be mapped on to time series of morphological change, illuminating the mode of acquisition of an adaptive complex.

The tempo and mode of evolution in response to environmental change and the role of behavioural innovation in this process[1] are critical for understanding the origin of adaptive traits. Understanding how functional relationships between traits and environmental variables emerged through time also underpins their use in predictive ecometric analysis. One of the most striking trends in the mammalian fossil record is the evolution of high tooth crowns (hypsodonty)[2-5]. Species across many orders of herbivorous mammals became hypsodont in apparent

[1]Department of Geosciences and Geography, University of Helsinki, Helsinki, Finland. [2]Natural History Museum, London, UK.
✉e-mail: juha.saarinen@helsinki.fi; a.lister@nhm.ac.uk

response to Late Miocene global shifts in climate and vegetation, which saw aridification and the spread of grasslands in Africa, Asia, the Americas and parts of the Mediterranean realm. Although this is generally considered to be a response to increased abrasion from feeding, it is debated whether abrasive plants (especially phytolith-rich grasses), or inorganic dust or grit ingested with food, are the principal drivers of the evolution of hypsodonty[6–13]. Furthermore, the evolutionary drivers of other dental traits, such as changes in numbers of cutting or shearing lophs and enamel thickness, are even less well understood.

Here, we focus on Proboscidea (elephants and relatives) from the Neogene to Quaternary of East Africa, presenting new data on the abundant fossil remains across a finely resolved stratigraphy that reveal dramatic changes in hypsodonty and other dental traits[14]. This is seen especially in the evolution of true elephants (Elephantidae), which arose from within a paraphyletic assemblage of 'gomphothere' proboscideans (Extended Data Fig. 1 and Supplementary Information Section 1) around 10–7 million years ago (Ma) in Africa. There, they differentiated into several genera including extant African elephants (*Loxodonta*) and Asian elephants (*Elephas*), as well as the extinct mammoths (*Mammuthus*) and straight-tusked elephants (*Palaeoloxodon*)[15], with parallel changes in dietary adaptation.

## Results and discussion

### Environmental change drove behavioural exploration of diet

To determine the pattern and causality of these changes (Figs. 1–4 and Extended Data Figs. 2–10), we compared trends in proboscidean dental traits (Figs. 3 and 4, Extended Data Figs. 6 and 7 and Supplementary Information Section 2) with a metric (dental mesowear angles[16]) that can be used to quantify the abrasion of enamel ridges on worn molar lophs/lophids (from here onwards called 'lophs') and directly assess the proportions of graze (grass and potentially other phytolith-rich herbaceous monocots) and browse (all other plant foods) in the diet[17–20] (Figs. 1 and 2; see Extended Data Fig. 9 for methodology).

We further compare dental morphometrics and mesowear with (1) direct palaeovegetation data from the proboscidean localities, (2) mean ordinated hypsodonty of the mammalian community as a proxy for local aridity and (3) terrigenous aeolian mineral dust flux from offshore cores as a measure of regional aridity and dust accumulation (Supplementary Data 2 and Supplementary Tables 1–3 in Supplementary Information Section 3).

Previous studies have used δ[13]C ratios as an index of the proportion of C3 and C4 plants in the environment and in animals' diet, generally taken as an index of browse versus grasses. This indicated that there was a dietary shift in various mammalian orders, including proboscideans, from browsing to grazing with the C3–C4 transition at around 10–8 Ma in East Africa[14,21]. However, C3 grasses, a potentially important dietary and selective factor, are invisible to this method; using dental mesowear, we demonstrate much earlier episodes of mixed to grass-dominated feeding within the C3-dominated assemblages of the earlier Neogene (Figs. 1 and 2, Extended Data Fig. 2 and Supplementary Data 1 and 2). Moreover, these episodes of dietary variation were closely linked to the vegetational environments in which the animals lived (Fig. 2 and Extended Data Fig. 3). Multiple regression commonality analysis (MRCA), with mesowear as the dependent variable, showed that the estimated grass percentage in fossil plant communities alone explains most of the variation in mesowear, whereas the effect of dust accumulation indicating aridification is not significant (Supplementary Table 3). Moreover, the diet of proboscideans varied spatially as well as temporally (Figs. 1 and 2, Extended Data Figs. 2–5 and Supplementary Information Section 4), and most of the variance, especially during the Late Miocene, was correlated with the estimated proportion of grass in fossil plant communities. This signals behavioural adaptation to local context, exemplified by the 'gomphothere' *Choerolophodon*, whose diet was graze-dominated at Fort Ternan, where pollen combined with stable isotope analysis indicate an abundance of C3 grasses as well as

sedges and reedmace (*Typha*)[22], then in the Ngorora Formation shifting to browse-dominated feeding in a forest environment in the Middle Miocene, followed by renewed grass consumption in the Late Miocene with a local transition from forest to grassy woodland (Fig. 2 and Extended Data Fig. 2). Spatial accommodation is shown at Moroto II, where the Early Miocene *Progomphotherium* grazed in a locally grass-rich habitat, in contrast to the generally forested environments of the time[23,24]. Furthermore, the presence of C4 grasses has now been documented in East Africa as early as the Early Miocene[24] (Fig. 2c). This in turn allowed flexible niche separation in mosaic environments, where locally open grass-rich habitats existed in otherwise forest-dominated context; for instance, at Maboko, where *Afrochoerodon kisumuensis* grazed in seasonal grassland whereas *Protanancus macinnesi* browsed in shrubland and/or woodland, as suggested by our mesowear data (Extended Data Fig. 2) and the palaeosol associations of these fossils[25]. Comparable results suggesting niche partitioning have been noted for Middle Miocene proboscidean communities in Europe[26] and Central Asia[27].

This intraspecific behavioural flexibility could have led to speciation of populations exploiting a new feeding niche. Crucially, similar behavioural 'experiments' are seen in the transition to C4 grass-feeding in the first true elephants (family Elephantidae). Thus, *Stegotetrabelodon* and *Primelephas* at Lothagam (Lake Turkana region, ca. 7.4–6.5 Ma) grazed in a locally grass-rich paleoenvironment, whereas at other East African localities ca. 6–5 Ma the same taxa show browse-dominated diets in a more wooded habitat (Fig. 2 and Extended Data Fig. 2). This prefigured the directional trend toward grazing specialization in elephants that began at ca. 4 Ma (Fig. 1). The C3–C4 transition during this period may[28] or may not[29] reflect an overall regional expansion of grassland, but our collation of local vegetational data show that from 5 Ma onwards, elephants occupied increasingly grassy areas (Fig. 2b). This led to an increase in the grass component of the diet (Figs. 1 and 2a), with potential selective pressure on dental morphology.

### Building of an adaptive complex

We mapped the evolution of major dental adaptations in proboscidean molars over 26 million years ago (Myr) (Fig. 3 and Supplementary Data 3–5). The quantified traits were hypsodonty, the number of enamel loops (lophs) and their spacing, enamel thickness and enamel folding, all of which have been broadly considered to be adaptations for resisting abrasion[30]. Early grazers *Afrochoerodon* (blue circles in Fig. 3) and *Progomphotherium* (blue 'Y' in Fig. 3) developed no clear dental adaptations to grass-eating, suggesting accommodation by behavioural means. Dietary flexibility in *Choerolophodon*, including mixed feeding in grassy biomes, is associated with thinner, more plicated enamel and slightly elevated hypsodonty in comparison with other 'gomphotheres'. These are minor compared with the drastic later shifts in elephants, but it is notable that at 12–10 Ma they predate the increase in dust accumulation[31] (Fig. 3 and Extended Data Fig. 6f), implying that diet is the selective factor.

The derived 'gomphothere' *Tetralophodon* is the probable sister-group of Elephantidae and is close to its ancestry[15], and samples dated to 10–9 Ma are taken as the starting point for subsequent morphological evolution. With third molars comprising 5–7 lophs, it was derived compared with earlier genera (3–4 lophs, exceptionally 5). Increased loph count reduces loph spacing (Extended Data Fig. 6) and is considered to enhance shearing efficiency[30]. This transition was facilitated by the flattening of the ovoid lophs into narrow, parallel-sided lamellae (Fig. 3g–i); these are associated with reorganization of the masticatory apparatus to allow proal (fore-to-aft) chewing at right-angles to the enamel bands, which has been considered to be the key innovation enabling further dental evolution in elephants[2]. We show, based on a test of phylogenetically correlated evolution, that the major increase in loph count in proboscideans was associated with and probably depended on the evolution of proal chewing (Supplementary Information Section 5). *Tetralophodon*, both at the

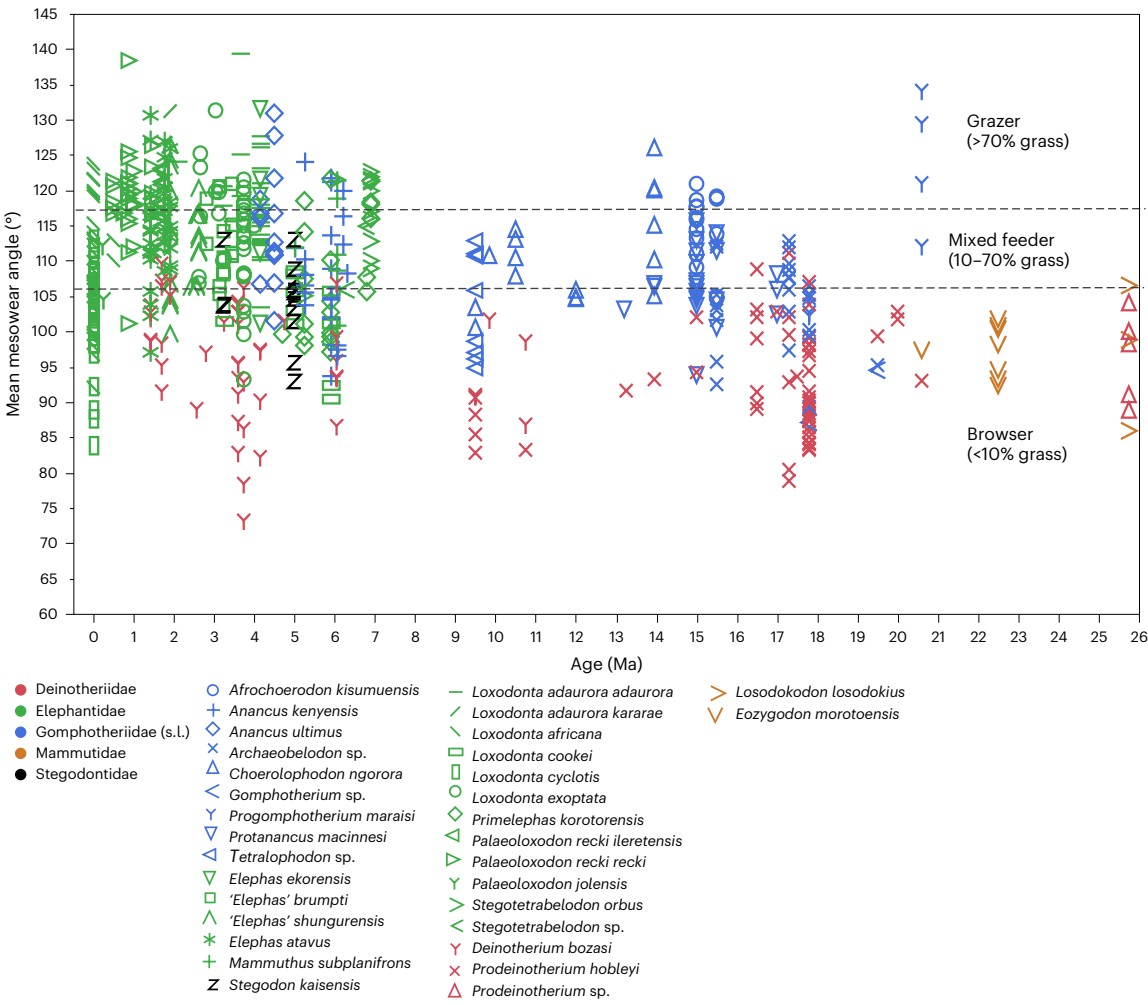

**Fig. 1 | Mesowear of East African Proboscidea 26 Ma to present.** Each data point corresponds to a molar. Dashed lines mark approximate thresholds between broad dietary categories (Methods). s.l. refers to sensu lato (in the broad sense).

localities studied here and at Nakali, Kenya (ca. 9.9 Ma), is considered to show the earliest evidence of proal chewing in the elephant stem group[2]. We found browse-dominated mesowear in most individuals, but a few were mixed feeders (Fig. 1), suggesting that the adaptations may have evolved as individuals exploited grassy areas in a largely wooded habitat[2]. Crucially, though, such changes enabled the later expansion of elephants into increasingly abundant grass-dominated environments and facilitated later dental changes driven by aridity and the association of the species with grassland.

The earliest elephants, *Primelephas* and *Stegotetrabelodon*, experimenting with grazing in the interval 7–5 Ma as discussed above, show a further increase to 6–9 lophs, as well as significant thinning of enamel and the beginnings of enamel folding, but not yet any increase in hypsodonty. The first increase in hypsodonty is seen in *Loxodonta cookei* at 6–5 Ma (significantly different from both *Primelephas* and *Stegotetrabelodon*), and the next increase in loph count is in *Loxodonta adaurora* at 4.2–4.1 Ma (significantly different from all three) (Supplementary Table 4). All dental characters subsequently underwent major directional change across several million years (Fig. 3 and Extended Data Figs. 6 and 10).

### Identifying environmental correlates of adaptive change

Experimental research has indicated that both dietary silica and exogenous dust affect rates of tooth wear[20,32–35]. To investigate which were the selective factors for the major trends in hypsodonty and other dental traits within the Elephantidae, we first conducted ordinary least squares (OLS) linear modelling (Supplementary Table 2) and MRCA (Supplementary Table 3), each with one trait as the dependent variable and inorganic abrasives (proxy for aridity) and dental mesowear (proxy for diet) as independent variables. Empirical and experimental data have indicated that mesowear reflects the amount of grass in the diet rather than the quantity of exogenous mineral dust, providing a means of separating the effects of these variables[17–20]. Although the mechanistic basis of this is uncertain, it has been suggested that it results from the large grain size of phytoliths compared with dust particles, or from grass leaves acting as a more rigid platform for phytoliths than for free-moving dust particles.

Our proxy for regional aridity was terrigenous dust accumulation in deep-sea sediment cores from site 659 off North-West Africa, which extends to 23 Ma and is the longest available record from anywhere in Africa[31,36], and from sites 721/722 in the Arabian Sea, which are closer to our study sites and span the past 7 Ma (ref. 37). Dust accumulation reflects regional aridity and shows a first-order increase through the past 7 Ma, with major fluctuations that have been linked to orbital forcing as well as high-latitude glacial–interglacial cycles after 2 Ma (refs. 38,39) (Fig. 4b and Extended Data Fig. 6f). As a proxy for local aridity, we used mean ordinated hypsodonty values of large mammal communities from the localities. Mean ordinated hypsodonty has been demonstrated to be a reliable proxy of precipitation today[40,41], and it can be used to reconstruct precipitation (and hence aridity) in

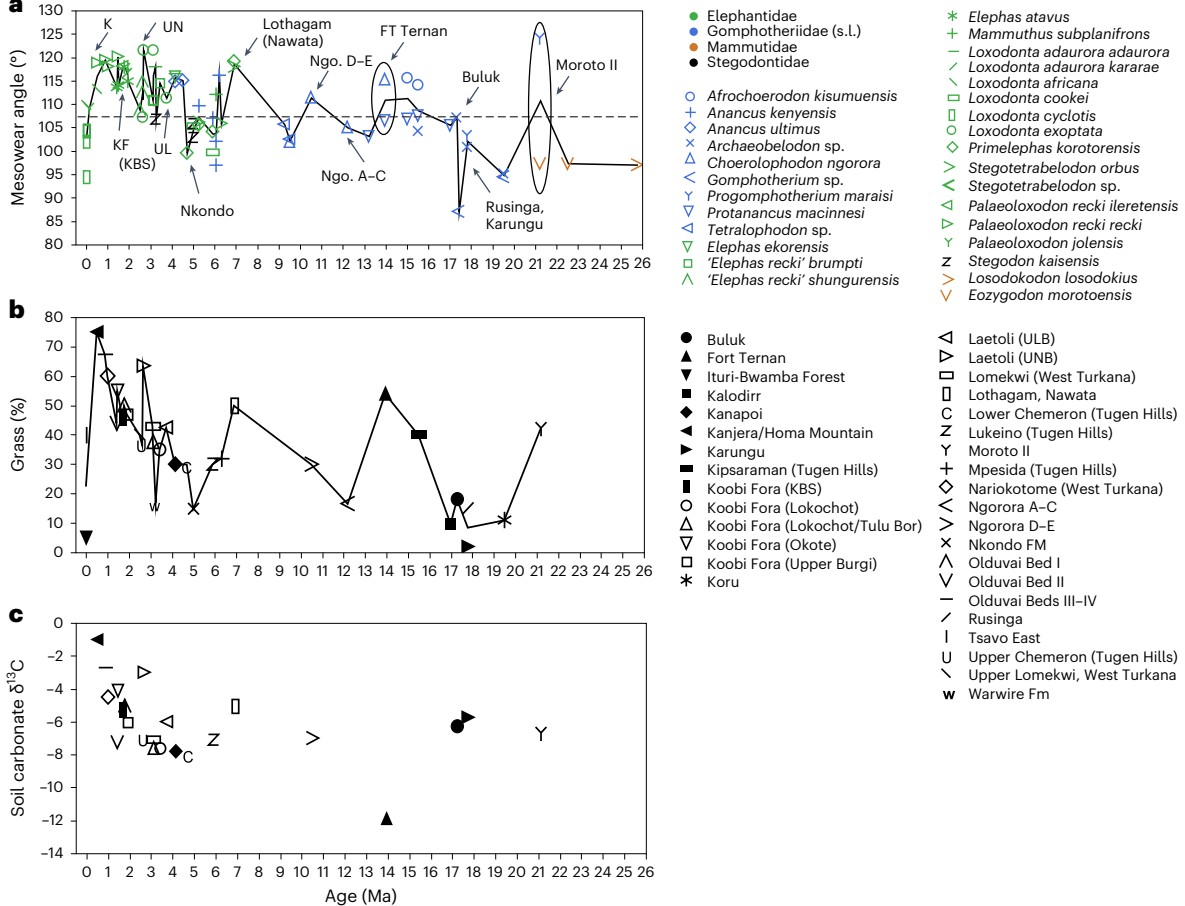

**Fig. 2 | Dietary variation of proboscidean populations in relation to vegetation. a–c**, Parallel variation in mesowear (dietary proxy) of proboscidean populations (**a**) and estimated grass cover of the locality (Supplementary Data 2) (**b**) compared with δ13C in carbonate nodules of sediments (**c**). Colours and symbols in **a** are as in Fig. 1. Example localities are indicated in **a** with arrows to highlight parallel variation between the mesowear and grass figures. Continuous lines connecting points are to aid visual comparison and do not necessarily indicate a temporal trend. The horizontal dashed line in **a** indicates the threshold between purely browsing mesowear angle values (below the line) and those indicating mixed feeding to grazing (above the line). Note early episodes of grass-dominated mixed feeding in *Choerolophodon ngorora* at Fort Ternan (ca. 14 Ma) and Members A–E of the Ngorora Fm (ca. 13.2–10.5 Ma). At 21 Ma, *Progomphotherium* from Moroto, Uganda, represents the earliest known evidence for a grass-dominated diet in a proboscidean. Note also the failure of the δ13C signal to identify C3 grassland, as at Ft Ternan. K, Kanjera; KF (KBS), Koobi Fora, KBS Member; UN, Laetoli Upper Ndolanya Beds; Nkondo, Nkondo Formation (Fm); Ngo D–E, Ngorora Fm Members D–E; Ngo A–C, Ngorora Fm members A–C; ULB, Upper Laetoli Beds; UNB, Upper Ndolanya Beds.

terrestrial paleoenvironments[11,42]. Comparison of dust accumulation with palaeovegetation has challenged the view that grassland expansion is primarily linked to aridity[31,43], and our MRCA results confirm stronger correlation of diet (mesowear) with grass cover estimates than with aridity proxies (dust and locality mean ordinated hypsodonty) (Supplementary Table 3). In addition, proxies indicate locally high grass cover at sites such as Moroto and Fort Ternan, when the regional climate was probably not arid. Following these observations, we treat aridity and grass cover as independent variables.

The OLS models show that hypsodonty, loph count, relative loph distance (loph distance/molar width) and enamel thickness have significant relationships with the aridity proxies (core 722 dust data and locality mean ordinated hypsodonty) but not with mesowear; this was valid whether considering all elephantoids or Elephantidae specifically over the past 7 Ma (Supplementary Table 2). Only enamel plication was significantly related to mesowear (as well as to regional dust data from core 722) but not to the local aridity proxy (mean ordinated hypsodonty), suggesting a functional response of this trait to grazing diet (Supplementary Table 2). Similar results were obtained for all elephantoids during the past 26 Ma, with plication showing an even stronger relationship with diet, being significantly associated

only with mesowear. There was also a relationship of hypsodonty and enamel thickness with mesowear, in addition to their relationship with local aridity.

We also ran MRCA over the past 5 Ma (the period of steepest directional change in dental morphology) to quantify the unique effects of mesowear and aridity on the dental traits (by removing potentially confounding correlations between them). This was done with both raw data and detrended data, to account for possible effects of autocorrelation. We found that local aridity (represented by locality mean ordinated hypsodonty) and diet, separately or jointly, explained ca. 35–60% of variation in major dental traits, whereas regional aridity and diet accounted for ca. 25–55% (Supplementary Table 3). Hypsodonty, loph count, relative loph distance and enamel thickness were predominantly related to the aridity proxies, confirming that they correlate principally with aridity of the environment, whereas enamel plication showed a mixed effect of aridity and mesowear.

These observations reflect functional differences across the dental traits. Hypsodonty increases the durability of the molar in the face of increased wear rate[4], and our results parallel observations in other large mammals, including suids[8] and equids[44], in suggesting increased aridity as the main driver of proboscidean hypsodonty. Further, our

results show that increase in loph count was primarily connected with the unique effects of aridification, suggesting that the primary selective advantage was durability and functional performance in arid environments. For enamel plication, on the other hand, both OLS analysis and MRCA indicated a relationship with mesowear, suggesting that plication would have been functionally beneficial for grazing elephantoids, probably because it increased the length of enamel bands and hence the shearing efficiency of the molars. A further proportion of variation in these traits was common to both aridity and mesowear proxies, leaving open the possibility of joint or synergistic causality.

These correlations across time and space suggest adaptation of the dental system to arid conditions and resulting increase in tooth wear. Although dust itself was probably the major selective force in these trends, plants living under arid conditions also contain more fibre (sclerenchyma)[45] and organic silica (phytoliths)[46], and C4 grasses have lower nutrient value than C3 grasses[47,48]; this would require greater food intake overall and more chewing, increasing lifetime abrasion and potentially contributing to selective pressure for the dental trends.

The thinning of the enamel bands that bound the lophs was also correlated only with aridity across the Elephantidae, with additional influence of grazing diet in the broader 26–0 Ma Elephantoidea analysis (Supplementary Tables 2 and 3). Enamel thickness, however, is linked both intraspecifically and interspecifically to the number and spacing of the lophs, corresponding to developmental coupling that became fixed at the species level, probably to maintain shearing efficiency[30] (Supplementary Data 6 and 7). The data do not support a model of enamel thinning as a driver of hypsodonty to compensate for reduced durability[4,5]. The frequency and amplitude of enamel folding (plication), however, are inversely correlated with enamel band thickness intraspecifically and interspecifically in Elephantidae, implying a possible developmental and functional link that was probably selected to maximize enamel volume and hence durability despite thinning. Nonetheless, we also show a relationship between plication and mesowear; this suggests that plication is associated with grazing diets and may have an adaptive role in increasing shearing efficiency by adding contact points between enamel edges as the lophs of the upper and lower molars slide against each other in a 'scissor-like' contact[49].

## The ratchet effect of stepwise evolution under varying climate

The correlation of dental traits with proxies for aridity and vegetation strongly suggests an adaptive basis for these traits. We used time-series analysis to identify these factors as the original drivers of change. The increase in hypsodonty and loph count in true elephants after 5 Ma was not only rapid but occurred in a stepwise fashion. Hypsodonty shows three major periods of increase (Fig. 4): (1) 5.0–3.75 Ma, culminating with *Loxodonta exoptata* at 3.74 Ma (range 3.8–3.5 Ma), *Elephas ekorensis* and *L. adaurora* at 4.15 Ma (4.2–4.1) and '*Elephas' brumpti* at 3.43 Ma (3.5–3.36 Ma); (2) 2.5–1.5 Ma, culminating with *Palaeoloxodon recki recki* at 1.46 Ma (1.53–1.38 Ma); (3) 0.5–0.13 Ma, culminating with *Palaeoloxodon jolensis* at 0.13 Ma ($n = 1$ only in East Africa, but elevated hypsodonty is corroborated in referred material from other regions of the continent[30]). The pattern of increasing loph/lamella numbers parallels the periods of increase in hypsodonty (Fig. 4), whereas the

decrease in enamel thickness was a more gradual process that started ca. 7 Ma after a shift to increasingly grazing diets (Extended Data Fig. 7).

Breakpoint analyses (Fig. 4) revealed a pattern where periods of rapid increase in hypsodonty and loph count (shaded areas in Fig. 4) alternate with longer periods of relative stasis in these traits. Moreover, the first appearance of each of these successively more dentally derived taxa corresponds to or slightly follows the three major episodes of elevated dust accumulation in the Arabian Sea record through 5–0 Ma (Fig. 4). This not only strongly corroborates aridity as a driver of dental evolution but illustrates the cumulative effect of successive pulses of change. Intervals of low aridity are accompanied by morphological stasis, not reversal. This indicates a ratchet effect, where the dental traits successively shifted to a new level with each increase in dust, remaining at that level until the next pulse pushed them to a further level (Extended Data Fig. 8). This model also explains the MRCA findings of lower correlation of morphological traits with regional compared to local aridity proxies, and the greatly reduced coefficients on removal of the first-order trend (detrending), as positive correlation at dust maxima is offset by negative correlation at dust minima.

The ratchet pattern occurred in the context of shifting species composition through the 7–0 Myr period, providing support for a neglected but important hypothesis of directional evolution. We suggest that the replacement of species through time in each genus is not simply the artificial subdivision of an anagenetic trend. Instead, it is likely to reflect the hypothetical process proposed by Futuyma[50] and Gould[51], in which ephemeral local adaptations contribute to long-term trends only when 'fixed' by speciation. Otherwise, they would be lost by continual range-shifting, splitting and merging at the population level. Hence, speciation facilitates directional evolution "by retaining, stepwise, the advances made in any one direction… Successive speciation events are the pitons attached to the slopes of an adaptive peak"[50]. Directional trends thus resulted in part from a form of population-level selection whereby only hypsodont populations or species survived episodes of extreme aridity.

Moreover, our findings of stepwise dental evolution in concert with peaks in selection pressure support the hypothesis that features such as hypsodonty are primarily adaptations to extreme rather than average conditions. The peaks in mean dust accumulation result mainly from an expansion of variation towards episodic high values (with medians remaining more constant)[31,37]. This provides an explanation for Liem's Paradox, in which mammalian species at a given time (for example, today) often have more specialized dentitions than their observed diets would predict[52]. Hypsodonty increases the longevity of molars and allows the consumption of abrasive foods without shortening lifespan[3]. Arising convergently in many mammalian lineages, it has hardly ever reversed to more brachydont crowns[3], implying that selection did not favour resource conservation in the face of reduced durability and developmental reorganization.

The adoption of C4 grazing in elephants occurred some time in the fossil-poor gap between 9 Ma (the last record of their browsing sister-group *Tetralophodon*) and 7 Ma (the first record of true elephants). The lag between the adoption of grazing between these dates and the onset of the major trend in hypsodonty and loph increase at 4 Ma led to the question of whether the dental change was adaptive to

**Fig. 3 | Trends in proboscidean dental traits in relation to aridity.**
**a–d,** Evolution of hypsodonty (**a**), number of lophs/lamellae (**b**), enamel thickness (mm) (**c**) and enamel plicae frequency (**d**) in East African Proboscidea 26–0 Ma. Symbols for species are as in Fig. 1; colours mark families or paraphyletic 'gomphothere' group (blue), with each icon representing one molar. Lines connect mean values of each taxonomic group per locality and are for visualization. **e-f,** Aridity proxies: mean ordinated hypsodonty values of proboscidean localities (locality symbols are as in Fig. 2) (**e**), aeolian dust accumulation data from cores 659 and 722 (**f**). **g–i,** Example photographs illustrating morphology and measurements of molars in lateral and occlusal

views: crown width (W) and height (CH) (hypsodonty = CH/W) (**g**), loph/lamella (L) and loph distance (LD) (**h**), and enamel thickness (ET) and plicae (PL) (plicae frequency = number of plicae in 1 cm of enamel band) (**i**). Left, *P. macinnesi* (Amebelodontidae) m3, Maboko, Kenya (NHMUK-PV-M15541a). Right, *P. recki recki* (Elephantidae) M3, Kanjera, Kenya (NHMUK-PV-M15418, reversed). Photographs are not to scale. The photographs on the left (*Protanancus*) represent a more plesiomorphic elephantoid molar and those on the right (*Palaeoloxodon*) represent a derived elephant molar. HYP, hypsodonty. Specimens photographed at the Natural History Museum, London.

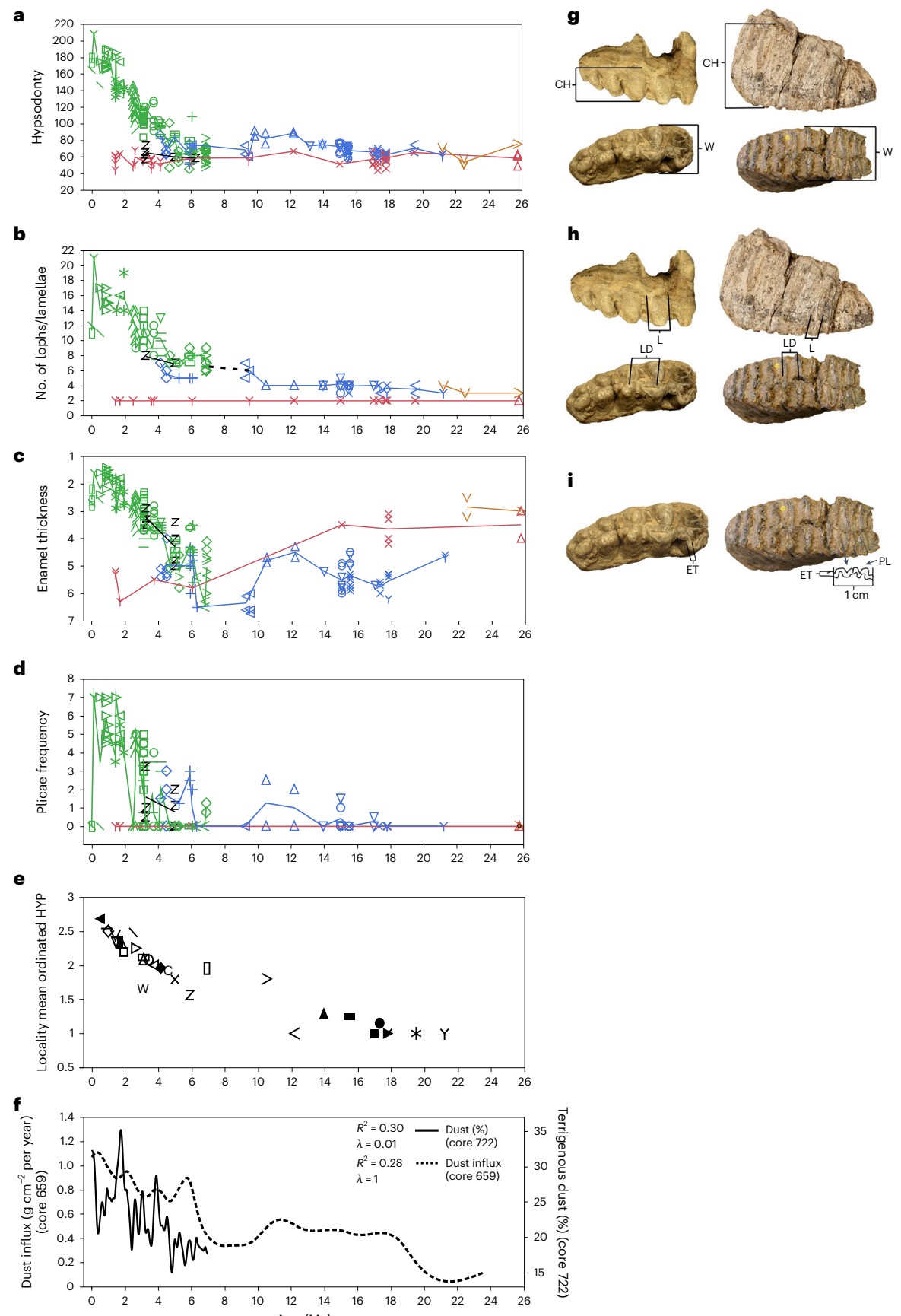

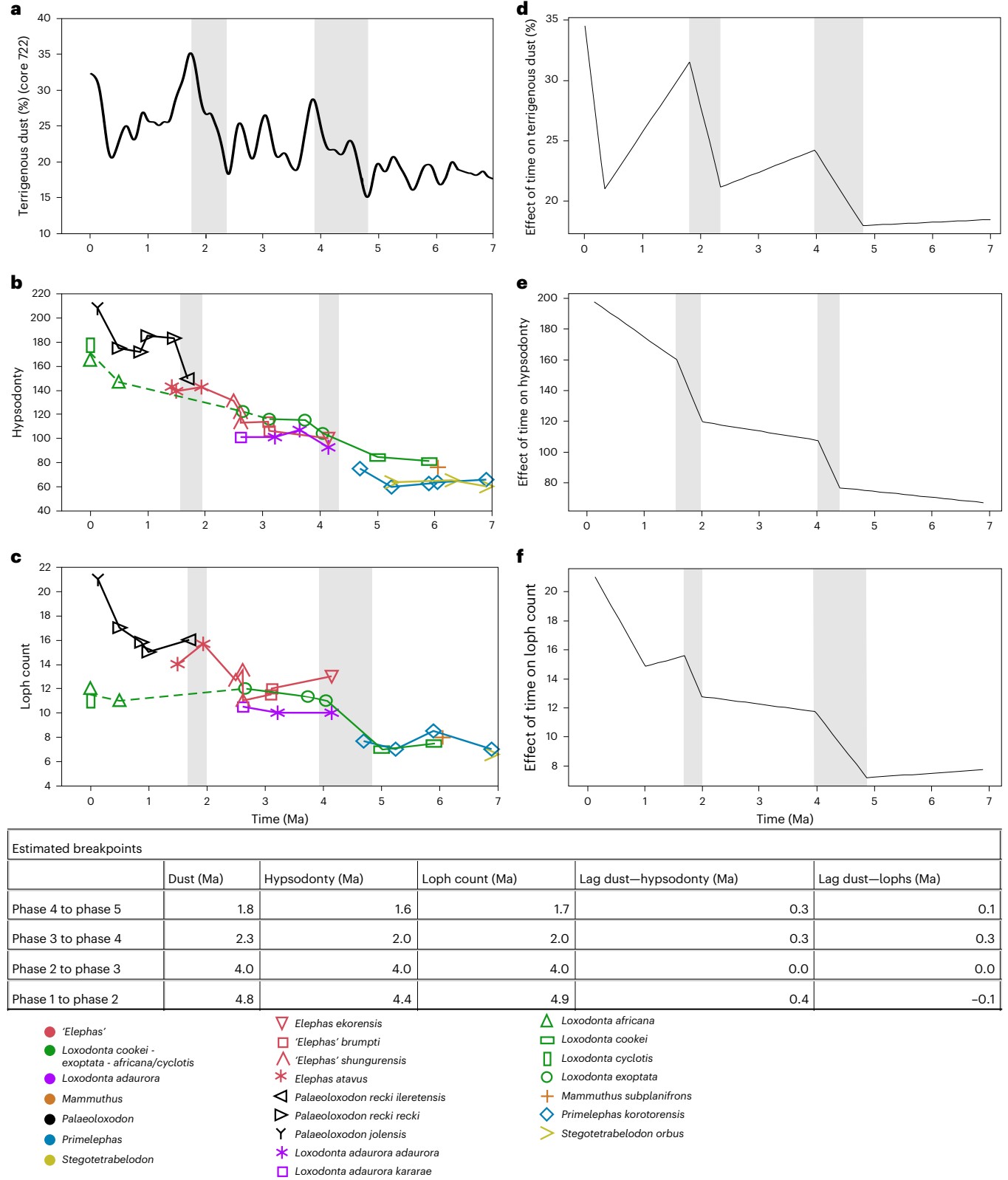

| Estimated breakpoints | | | | | |
|---|---|---|---|---|---|
| | Dust (Ma) | Hypsodonty (Ma) | Loph count (Ma) | Lag dust—hypsodonty (Ma) | Lag dust—lophs (Ma) |
| Phase 4 to phase 5 | 1.8 | 1.6 | 1.7 | 0.3 | 0.1 |
| Phase 3 to phase 4 | 2.3 | 2.0 | 2.0 | 0.3 | 0.3 |
| Phase 2 to phase 3 | 4.0 | 4.0 | 4.0 | 0.0 | 0.0 |
| Phase 1 to phase 2 | 4.8 | 4.4 | 4.9 | 0.4 | −0.1 |

**Legend:**
- 'Elephas'
- Loxodonta cookei - exoptata - africana/cyclotis
- Loxodonta adaurora
- Mammuthus
- Palaeoloxodon
- Primelephas
- Stegotetrabelodon
- Elephas ekorensis
- 'Elephas' brumpti
- 'Elephas' shungurensis
- Elephas atavus
- Palaeoloxodon recki ileretensis
- Palaeoloxodon recki recki
- Palaeoloxodon jolensis
- Loxodonta adaurora adaurora
- Loxodonta adaurora kararae
- Loxodonta africana
- Loxodonta cookei
- Loxodonta cyclotis
- Loxodonta exoptata
- Mammuthus subplanifrons
- Primelephas korotorensis
- Stegotetrabelodon orbus

**Fig. 4 | Ratcheted hypsodonty and loph count increase in Elephantidae and dust flow record 7–0 Ma as indicated by breakpoint analyses. a**, Terrigenous dust percentage in ODP cores 721/2. **b**, Hypsodonty of Elephantidae. **c**, Loph count of Elephantidae. **d–f**, Breakpoint analysis for **a–c**, respectively. Grey bars indicate times of increase in each variable delimited by estimated breakpoints in **d–f**, transferred to **a–c**. Both hypsodonty and loph count increases occur during times of major surge in dust flow, 4.8–4 Ma and 2.3–1.8 Ma, as delimited by estimated breakpoints, with no or short lag. A third coincident breakpoint in dust

and dental variables is probable between 0.5–0 Ma, but this was not recognized by the algorithm owing to a lack of data beyond the present. Stasis is seen 7–5 Ma in *Stegotetrabelodon orbus*, *Primelephas korotorensis* and *L. cookei*; 4–2.5 Ma in *L. adaurora*, *L. exoptata* and *Elephas* spp; and 1.5–0.5 Ma in *P. recki*. Lines in **b,c** are for visualization and do not necessarily indicate direct ancestry. *L. exoptata* is considered to be close to the ancestry of *L. africana*[15], but the pattern of change is unknown 2.5–0.5 Ma, indicated by the dashed green line. The dust data are represented by a smoothing spline curve with $R^2 = 0.3$ and $\lambda = 0.015$.

grass-eating but lagged behind the behavioural trigger or was adaptive to another selective force concurrent with it[14]. The data presented here suggest that the secular increase in aridity and airborne dust was the principal driver of the adaptive trend in abrasion resistance. Moreover, small morphological advances 7–5 Ma are coincident with the beginning of the aridity trend. Lags between the adoption of grazing and the evolution of hypsodonty have been observed in mammalian guilds on other continents[53,54], with hypsodonty suggested as a response to aridity or grit ingestion[55,56].

Nonetheless, behavioural experimentation with grazing (a form of phenotypic plasticity that can presage evolutionary change[57]) set the context for morphological adaptation, as early elephants explored more open environments. Following the innovations of proal chewing and lamelliform lophs around 10 Ma, loph numbers first increased in the elephantid precursor *Tetralophodon*, a probable response to feeding on low-growing vegetation and the consequent incorporation of grit even at low levels of airborne dust. A shift to grazing and increased aridity beginning around 6 Ma then drove a further increase in loph count and the first reduction in enamel thickness (Lothagam *Stegotetrabelodon* and *Primelephas*, dated in the range 7.4–5.0 Ma; Extended Data Fig. 6a–c). The first major peak in dust flux in the North-West African core at 6 Ma (Fig. 3f) also corresponds to the first elephant with elevated hypsodonty (*L. cookei*). It is matched by only a minor peak in the Arabian Sea core (Fig. 3f), but a major episode of aridity at this time in East Africa is indicated by pollen spectra from a nearby core[43]. Aridity and the prevalence of grass-eating increased further from ca. 4 Ma, the former leading to the major, ratcheted trends in hypsodonty, loph count and packing, and enamel thinning. Enamel plication is the trait most clearly associated with grazing diets, enhancing shearing efficiency by increasing the length of enamel bands.

The ratchet pattern observed here for proboscidean hypsodonty and loph count may have generality across other mammalian orders and further traits increasing dental durability. Madden[8] plotted molar occlusal surface areas of two suid and two primate species from East African sites in the interval 4.5–1.5 Ma, suggesting that increases coincided with peaks in an offshore dust core, although the relation was not tested statistically or in comparison with dietary or vegetation proxies.

### Diversification and extinction

The pattern of morphological change among African Proboscidea as a whole is one of increasing disparity upward from a continuing baseline of low loph count (two being the minimum possible) and low crowns, maintained throughout by the browsing deinotheres (Fig. 3, red symbols). Moreover, early experiments in grazing within 'gomphotheres' did not lead to substantive dental specialization. Grazing on relatively high-nutrient C3 grasses in a humid environment favouring less fibrous species, together with a low prevalence of dust, presumably maintained low selection pressure on dental traits. Morphological constraints may also have played a part: the multiplication of lophs could not commence until the 'invention' of proal chewing in the precursors of true elephants, whereas the development of hypsodonty may have been developmentally inhibited by morphological constraints such as bunodonty and thick enamel. The acquisition of proal chewing in stegodonts in Asia (before their dispersal to Africa), in parallel with the elephants, led to lamellar multiplication but not to hypsodonty increase[2]. In Africa, the last-surviving 'gomphothere' *Anancus* underwent only minor parallel changes: an increase in last molar loph count by one or two lophs and development of some enamel plication (Fig. 3 and Extended Data Fig. 6).

The expansion of C4 grasslands from 10 Ma and the adaptation of true elephants to grazing saw a gradual shrinking of proboscidean diversity, with the extinction of 'gomphotheres' (*Anancus*) at around 4 Ma, stegodonts (*Stegodon*) at 3 Ma and deinotheres (*Deinotherium*) at 1.5 Ma (ref. 58). This formed part of a general decline in megaherbivore browsers and mixed feeders with the expansion of grasslands[59]. The latest-surviving proboscidean taxa in Africa, into the Late Pleistocene,

were the extreme grazing-adapted elephant *P. jolensis* and the last representatives of *Loxodonta*, the extant African elephants *Loxodonta africana* and *Loxodonta cyclotis* (the latter not represented in the known fossil record). Extreme aridity in the second half of the Pleistocene after ca. 1 Ma appears to have reversed this trend, expanding shrubby semidesert and disfavouring grazers[60]. The dental trends of *Loxodonta* progressed in parallel with those of other genera but to a more moderate extent (Fig. 4 and Extended Data Fig. 7), reflecting a more generalized diet than that of the grazing *Palaeoloxodon* that perhaps explains the sole survival of *Loxodonta* among African Proboscidea. Today's African elephants are browsers and mixed feeders (Fig. 1) inhabiting forests, savanna grasslands and desert shrublands, their broad niche facilitated by the absence of proboscidean competitors.

## Methods

The fossil and extant proboscidean material studied here is conserved at the Natural History Museum, London, UK (NHMUK); Museum für Naturkunde, Berlin, Germany; Royal Museum of Central Africa, Tervuren, Belgium; Tsavo Research Station, Tsavo East National Park, Kenya; the National Museums of Kenya, Nairobi, Kenya; Uganda Museum, Kampala, Uganda; and National Museum of Tanzania, Dar es Salaam, Tanzania. Altogether, ca. 500 molars of proboscideans from the past 26 Ma were studied for mesowear and morphometrics, including 32 modern savanna elephant *L. africana* and 43 forest elephant *L. cyclotis* specimens (Supplementary Data 1–5). Taxonomic issues relating to the samples are discussed in Supplementary Information Section 1.

### Mesowear angle measurements and dietary categories

The method used to measure mesowear angles of proboscidean molars followed the procedure introduced by Saarinen and colleagues[16], extended to cover buno-lophodont 'gomphothere' molars and facet slope-based mesowear angles (Extended Data Fig. 9a–c). All sufficiently preserved molariform teeth apart from dP2/dp2 and dP3/dp3 were scored (Supplementary Data 1). For buno-lophodont molars, the mesowear angle was measured from the deepest dentine valley within a loph, which corresponds with the practice of measuring from the middle of worn dentine valleys in elephant molars. We demonstrate that mesowear angles measured from facet slopes can be used to complement the mesowear data from dentine valleys, as the difference between the mean facet and dentine valley angles is negligible and non-significant in deinotheres that have consistently similar browsing dietary composition (Extended Data Fig. 9d). Moreover, mesowear angles measured from dentine valleys and facet slopes have been shown to be the same within worn 'gomphothere' molars[61]. We suggest this consistency between facet-based angles and mesowear angles measured from worn dentine pits and/or valleys is due to processes that maintain similar mean slopes across worn occlusal surfaces with similar wear stages and similar dietary composition[62]. Comparisons of the mean slope of worn enamel features at occlusal surfaces between five wear stages in five extant primate species have shown that with the exception of nearly unworn and extremely worn molars, the mean slope of occlusal surfaces remains similar in different wear stages within species, with differences mainly between species following dietary differences[62].

Mesowear data are graphed either by individual specimen (Fig. 1) or as species means per site (Fig. 2a). We followed the principle of Hoffman and Stewart's definition[63] of dietary categories as browsing (less than 10% grass in diet), grazing (more than 90% grass in diet) and mixed feeding (10–90% grass in diet). However, our thresholds between broad dietary categories were based on regressions between mesowear and stable carbon isotope proxies of diet in East African proboscideans[16], and owing to a possible effect from C3 as well as C4 grasses we set the grazer signal at values indicating more than 70% C4 plants. Moreover, the threshold mesowear angle marking ~70% grass in diet has been modified from 124° to 117° based on a revised mean of mesowear angles

corresponding to δ13C > −2‰ (Supplementary Data 2). Species and/or site means are used in statistical analyses.

## Dental morphometrics

Morphometric data were collected from last molars (M3 and m3) for elephantoids (Mammutidae, 'gomphothere' families, Stegodontidae and Elephantidae), whereas for Deinotheriidae, second molars (M2 and m2) were included to enhance sample sizes, because of the structural and morphological similarity of second and third molars in deinotheres. Some M2/m2 data, for enamel thickness only, were also included for *Tetralophodon* to increase the sample size. Proboscidean molar morphology is discussed in Supplementary Information Section 2. The measurement protocol is shown in Fig. 3e and is based on those of Lister and Sher[64] and Beden[65]. Molar crown width was measured as the greatest width along the molar crown, including cement. Crown length is the greatest length of the crown normal to the average orientation of lamellae. Crown height was measured as the height of the anteriormost unworn loph/lamella in the molar from the bottom to the top of the crown (this measurement was only taken from unworn and moderately worn molars on unworn lophs/lamellae). Hypsodonty was calculated as: (crown height/crown width) × 100. Numbers of lophs/lamellae were counted only for complete tooth crowns that had not lost lamellae, excluding anterior and posterior cingulae ('talons'). The distance between lophs/lamellae was measured from the midline of a loph/lamella to the midline of the following loph/lamella and averaged across all pairs of adjacent lophs/lamellae. Enamel thickness was measured parallel to the walls of an enamel ridge exposed by wear (not along inclined enamel surfaces). Enamel thickness was measured at three points on the molar surface (where possible) and averaged. Frequency of enamel plications (plicae frequency) was counted as the number of enamel folds in a 1 cm length of a fully exposed enamel ridge on the worn occlusal surface of the molar. Plicae frequency was also measured at three points on the worn occlusal surface (when possible) and averaged. Plicae amplitude was measured as the distance between three upper and lower peaks of plication in the enamel band. Where possible, this was taken at three points within the enamel band and averaged. The dental morphometric data can be found in Supplementary Data 2–5.

## Patterns of dental trait evolution in relation to phylogeny and taxonomy

We interpret the successive appearances of more derived morphology with the Elephantidae, interspersed with episodes of stasis (Figs. 3 and 4 and Extended Data Figs. 6, 7 and 10), as reflecting a true evolutionary pattern despite the uncertainty of the precise relationships among the species and subspecies in the sequence. The first appearance of each morphotype in East Africa generally represents its global FAD (first appearance datum)[15]; indeed Fortelius et al.[42] suggested that the Turkana region, with early aridification, probably acted as a 'species factory' where many groups of mammals first started to adapt to expanding grasslands and seasonally dry climatic conditions. Hence, there is no evidence of immigration masquerading as in situ evolution, even where the immediate ancestor of a newly appearing morphotype is unidentified. Moreover, trait evolution was remarkably parallel among genera (Extended Data Fig. 7); that is, shifting the generic attribution of a species would not affect the overall pattern. Finally, episodes of stasis within species are robust irrespective of generic assignment. In the figures, lines connecting chronologically successive samples are to aid visualization and do not necessarily represent direct lines of descent.

We tested the hypothesis that major increases in loph count in proboscideans (beyond five in last molars) were facilitated by the evolution of a propalineal (proal)—that is, fore–aft—chewing cycle by performing Pagel's test[66] for the global proboscidean supertree of 185 species of proboscideans presented by Cantalapiedra et al.[58]. For comparison, we similarly tested whether the evolution of hypsodonty in proboscideans was phylogenetically correlated with the evolution of proal chewing.

The trait data used in the analysis are provided in Supplementary Data 8. The results are presented in Supplementary Information Section 5 (and Supplementary Tables 5 and 6 there).

## Stratigraphy and palaeoenvironments

The stratigraphic age of the proboscidean specimens was based mainly on radiometrically dated volcanic tuffs[67–70] (Supplementary Data 2–4). Where a range is given, the median age is used in graphing and analyses. Estimates of grass percentages of the vegetation in paleoenvironments are based mainly on published pollen and plant macrofossil records, with additional information from mammalian and molluscan assemblages, phytoliths and soil carbonate δ13C records (Supplementary Information Section 6). We acknowledge the most recently published estimates of presence of C4 photosynthesizing grasses in the Early Miocene[24]. However, because of the non-analogue nature and highly heterogenous signal of vegetation proxies for the Early Miocene localities, and the fact that some of the evidence for abundant C4 grasses in Rusinga, for example, does not come from the same level as the proboscidean fossils, we collated a combination of studies and proxies for the estimation of grass percentage of the vegetation for Early Miocene localities (Supplementary Information Section 6). The relationship of grass percentage with soil and enamel carbon isotope records, especially in relation to the role of C3 grazing, is further explored in Supplementary Information Section 4.

The record of aeolian (terrigenous) dust accumulation during the past 7 Ma is based on deep-sea sediment cores from the Arabian Sea[37] and North-West Africa[31,36], which reflect the influx of aeolian dust from the horn of Africa and the north-west Sahara, respectively. The Arabian Sea record (Ocean Drilling Program (ODP) Sites 721 and 722) of terrigenous dust percentage, spanning ca. 8 Myr, is close to our study area and is therefore used in the OLS multiple linear regression model and MRCA (see below). The bottom part of the core (7–8 Ma) is, however, considered to be unreliable (P. deMenocal, personal communication) and is therefore excluded. The West Atlantic record (ODP Site 659) of dust flux is corrected for sedimentation rate and dry density[31] and extends to ca. 23 Ma, so it was consulted for the first-order pattern across the whole time interval, given evidence that major patterns of aridification are broadly continent-wide[38,71]. We took raw data from the source references and fitted a smoothing spline curve for graphing, maintaining $R^2$ values of ca. 30% in the interpolation of the raw data. For OLS analysis and MRCA, we averaged dust percentage or flux values across the age range of the sample (Supplementary Data 2). Locality mean ordinated hypsodonty values were obtained from the NOW-database (data (https://nowdatabase.org/now/database/) by the NOW Community / CC BY 4.0), and from additional literature sources (see Supplementary Data 2).

Extended Data Fig. 10 presents a summary of all the essential data used in the analyses for visual inspection and comparison.

## Statistical analyses

Dietary differences among proboscidean species and populations (Extended Data Fig. 2) were tested with pairwise Wilcoxon tests on the sample means of mean mesowear angles using SAS JMP Pro 14.

We used an OLS approach for regressions between the time series of dental traits, dietary proxies (mesowear) and aridity proxies (locality mean ordinated hypsodonty and dust accumulation record from the marine sediment cores). The potential effect of autocorrelation (in particular, the autoregression—that is, the autocorrelation of residuals—between time series) should be accounted for in analyses of time series. As a first step, therefore, we tested for autoregression by performing Durbin–Watson tests[72] for the complete elephantoid data from 26 to 0.13 Ma, using R package 'car' in RStudio v.3.5.3. The Durbin–Watson tests, which compare adjacent data points in a time series, did not indicate significant autoregression for any of the dental traits in a multiple regression model that had mesowear, core 722 and

core 659 dust data and locality mean ordinated hypsodonty as the independent variables (Supplementary Table 1). Based on the lack of autoregression, the correct models to choose in these cases were OLS rather than autoregressive-moving average models[73,74]. Note that in the OLS results (Supplementary Table 2), significant values have negative signs for loph distance/crown width and for enamel thickness; this is because these variables reduce in value in response to an abrasive diet (lophs pack more closely and enamel becomes thinner), whereas hypsodonty and loph number increase.

To explore the correlates (putative causal factors) of dietary change, climate and dental morphology, we used MRCA[75,76], implemented in IBM SPSS Statistics v.25. This method partitions variance into that explained uniquely by each of the predictor (independent) variables and that which is explained by them in common but cannot be uniquely allocated. The method was used to explore (1) how much of the variation in mesowear was explained by terrigenous percentage or estimated grass percentage and their 'common' (that is, indivisible) effects; and (2) how much variation in dental metrics was explained by terrigenous percentage, locality mean ordinated hypsodonty, mesowear and their common effects. The mesowear analysis was run across all proboscidean populations excluding deinotheres (means extracted from Supplementary Data 2, $n = 81$); modern African elephants were excluded because their habitats, and therefore probably diets, have been influenced by anthropogenic factors[77]; 0.13 Ma is the age of the youngest fossil data point, *P. jolensis* from Natodomeri. The dental trait analysis was run on all elephantid populations from 5 to 0 Ma (including modern African elephants); means were extracted from Supplementary Data 4, $n = 41$. To account for possible effects of autocorrelation, we performed the analyses with detrended data (data with linear trends through time removed) as well as the original data. The detrending reduced the overall robustness of the models drastically but proportional unique versus common effects of mesowear and the aridity proxies (locality mean ordinated hypsodonty and terrigenous dust) on dental traits were mostly retained; however, the unique effect of mesowear on loph packing (loph distance/molar width) and enamel thickness increased (Supplementary Table 3). Note that negative coefficients in MRCA indicate that the variable is acting to 'suppress' the effect of another and are subtracted in the calculation of total explained variation[75].

Interrelationships among dental parameters are strongly collinear across species and populations (Supplementary Data 9 and 10), so to test for probable functional links we used partial correlation analysis (implemented in Statistica 13.3, Tibco). Analyses were run separately across all elephantid population means (extracted from Supplementary Data 4, $n = 41$) and all named species or subspecies means (Supplementary Data 5, $n = 16$) with crown height, loph/lamella count, lamellar frequency, enamel thickness, plicae frequency and plicae amplitude as variables. In each run, the partial correlation of one pair of variables was obtained, with the other variables kept constant (Supplementary Data 6). Crown width, as a measure of molar size, was included as a covariate in partial correlation analyses. We therefore used crown height rather than hypsodonty in these analyses as the latter (hypsodonty = crown height/crown width) is already corrected for crown width. Similarly, lamellar frequency, an index of the spacing between lamellae or lophs, is influenced by both loph count and (inversely) molar size[78]. Partial correlation analyses were therefore run both with and without crown width as a covariate, in the latter case allowing lamellar frequency to reflect the absolute spacing between lophs (and hence enamel bands), of probable functional significance irrespective of molar size.

Correlations within species are likely to reflect developmental links, but too few specimens preserve all measurements for partial correlation analysis, so product-moment correlation coefficients were obtained for each pairwise combination of variables; this yielded a limited, meaningful set of significant correlations (Supplementary Data 7). For bivariate intraspecific correlations, we analysed both

lamellar frequency, and lamellar frequency × crown width against other variables; the latter effectively corrected for the inverse relation of lamellar frequency to crown size.

$P < 0.05$ was considered to indicate statistical significance, but we also note results where $0.05 < P < 0.10$ as potentially significant, especially where sample sizes were low[79].

Finally, we tested a hypothesis of a 'ratchet effect' of the regional proxy of climatic aridification (core 722 terrigenous dust percentages) on dental traits using breakpoint analyses and broken-line regressions with the R package 'segmented' in RStudio v.3.5.3 (ref. [80]). Significant breakpoints were identified in the core 722 terrigenous dust percentage data and in hypsodonty and loph counts of elephants, and we compared these patterns to identify a ratchet effect between aridification (dust accumulation peak) events and incremental increase in hypsodonty and loph count. The most recent (Middle Pleistocene to extant) *Loxodonta* were excluded from the breakpoint analyses because of the long gap in the occurrence of this genus in the East African fossil record between ca. 2.5 and 0.5 Ma and the subsequent uncertainty about the area of occurrence and habitats of *Loxodonta* during this time.

### Reporting summary
Further information on research design is available in the Nature Portfolio Reporting Summary linked to this article.

## Data availability
All original data used in the study are included in Supplementary Data 1–10 and in the figshare online repository (https://doi.org/10.6084/m9.figshare.23276126). The proboscidean supertree data used in Pagel's test (Supplementary Information Section 5) has been previously published by Cantalapiedra et al.[58].

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

## Acknowledgements

For access to proboscidean collections, we thank P. Brewer and R. Pappa (NHMUK); T. Schossleitner and O. Hampe (Museum für Naturkunde, Berlin, Germany); F. Mees and E. Gilissen (Central Africa Museum, Tervuren, Belgium); J. Kibii, F. K. Manthi, R. Nyaboke, P. Mbatha, J. Yatich and F. Ndiritu (National Museums of Kenya, Nairobi); J. Kipchere (Kipsaraman Museum, Kipsaraman, Kenya); S. Ngene (Tsavo Research Station, Voi, Kenya); A. Mugume (Uganda Museum, Kampala); and A. Kidna and G. Kamatula (National Museum of Tanzania, Dar es Salaam). NACOSTI, Kenya, COSTECH, Tanzania, and Uganda Museum (Kampala), Uganda, granted permits for the research, and further assistance with materials from Uganda was provided by L. MacLatchy, S. Cote and M. Pickford. We also thank W. Sanders and H. Zhang for discussions on proboscidean taxonomy; J. Cantalapiedra for discussions on testing phylogenetic correlations of traits; T. Reitan for discussions on time-series causality analyses; S. Sova, I. Žliobaitė and M. Fortelius for discussions on dental traits; and P. deMenocal on dust records.

## Author contributions

J.S. and A.M.L. conceived the project. J.S. collected the data and prepared graphics. J.S. and A.M.L. performed statistical analyses and wrote the paper.

## Funding

 Work by J.S. was funded by the Jenny and Antti Wihuri Foundation and by the University of Helsinki and Academy of Finland (project numbers 315691 and 340775/346292).

## Competing interests

The authors declare no competing interests.

## Additional information

**Extended data** is available for this paper at https://doi.org/10.1038/s41559-023-02151-4.

**Correspondence and requests for materials** should be addressed to Juha Saarinen or Adrian M. Lister.

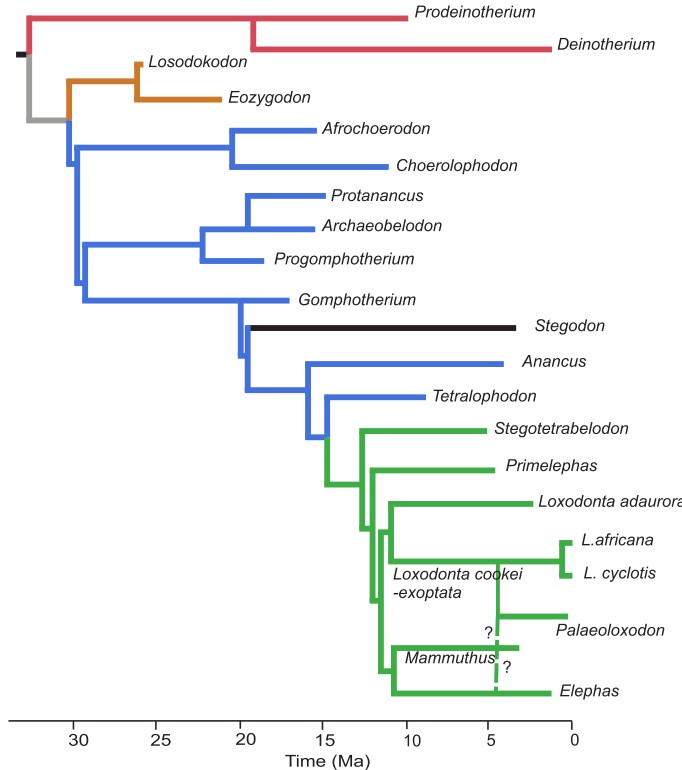

**Extended Data Fig. 1 | Cladogram of East African Proboscidea from the last 26 Ma.** Deinotheriidae is indicated in red colour, basal stem of Elephantoidea in grey, Mammutidae in brown, Stegodontidae in black and Elephantidae in green. The families shown in blue are often grouped informally as 'gomphotheres'. Nodes indicate relationships but not precise divergence ages. The cladogram is based on the Proboscidean supertree of Cantalapiedra et al.[58]. The dashed line from *Elephas/Mammuthus* to *Palaeoloxodon* indicates a suggested influence of hybridization into *Palaeoloxodon* from those lineages (see references in Supplementary Information 1).

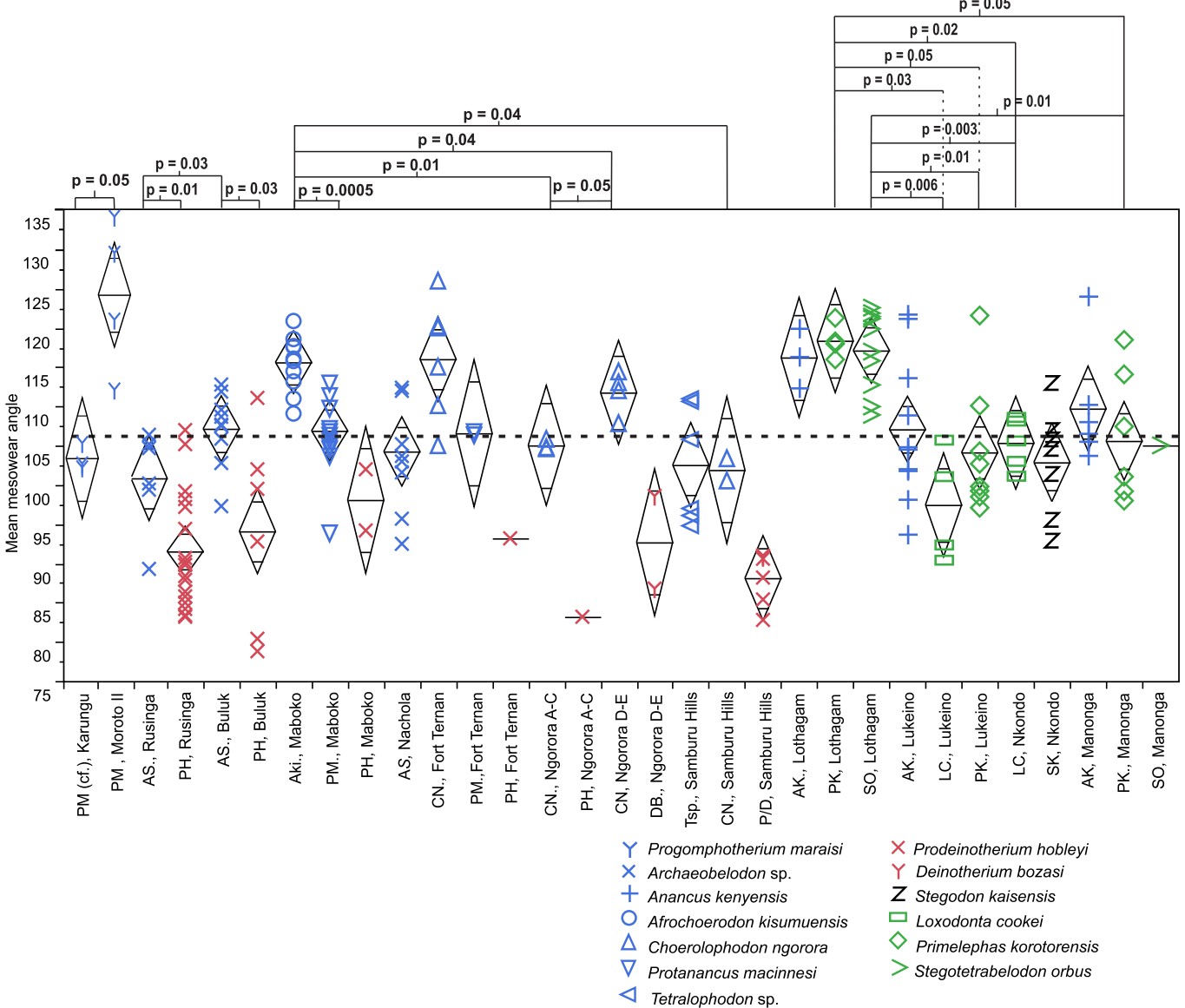

**Extended Data Fig. 2 | Mean mesowear angles of selected proboscidean populations from East African Miocene localities.** Mean mesowear angles of selected proboscidean populations from East African Miocene localities, with two-sided pairwise Wilcoxon tests. Symbols show mean mesowear angle of each specimen; diamonds show population mean (central line) and 95% confidence interval (upper & lower line). Localities arranged left to right in approximate order from oldest to youngest. The dashed line indicates tentative threshold value of mean mesowear angle (106°) separating browsers (<10% grass in diet) from mixed-feeders and grazers ( > 10% grass in diet). Statistically significantly different p-values are indicated for selected population pairs.

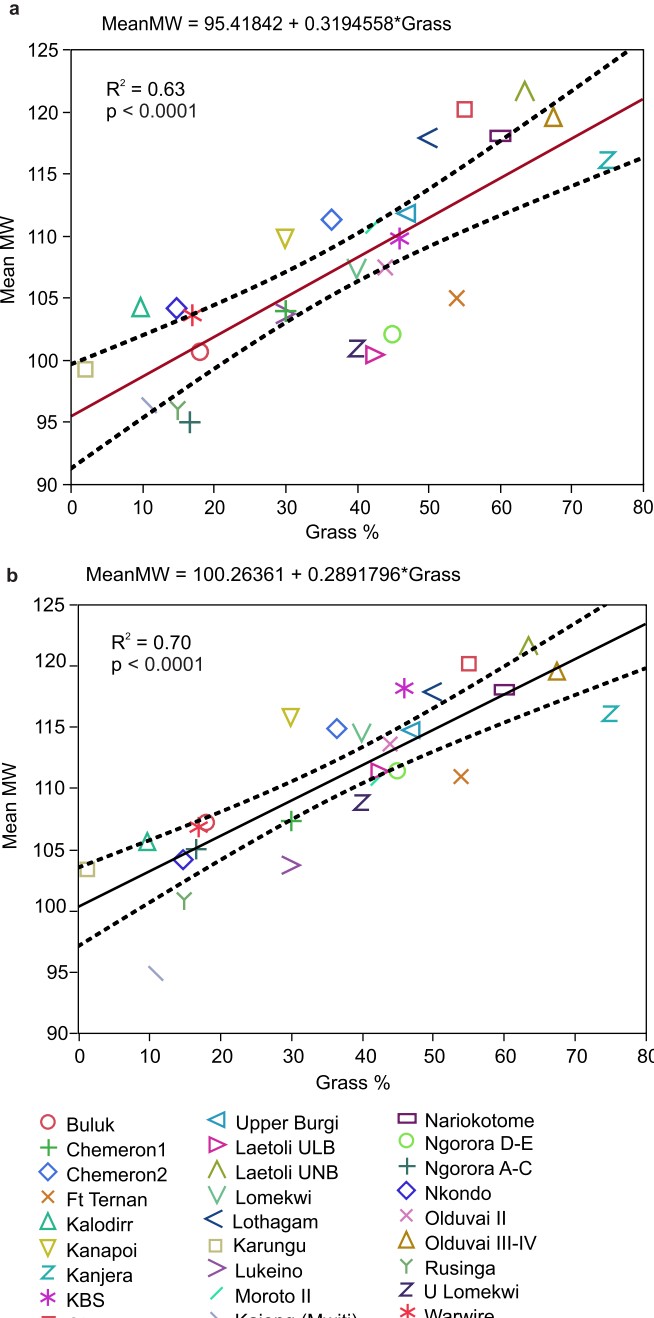

**Extended Data Fig. 3 | Ordinary least squares regressions between estimated percentage of grass cover and mean mesowear angle of proboscidean guild within localities. a)** Mean mesowear angle of all proboscideans (including deinotheres) within localities ($n = 27$, $R^2 = 0.63$, $p = 0.0000008$) and **b)** mean mesowear angle of elephantoid proboscideans only, that is excluding deinotheres ($n = 27$, $R^2 = 0.70$, $p = 0.00000004$). Both show significant positive correlations between mean mesowear of proboscideans and estimated grass percentage. 95% confidence limits of fit are indicated as dashed curves in figures a and b.

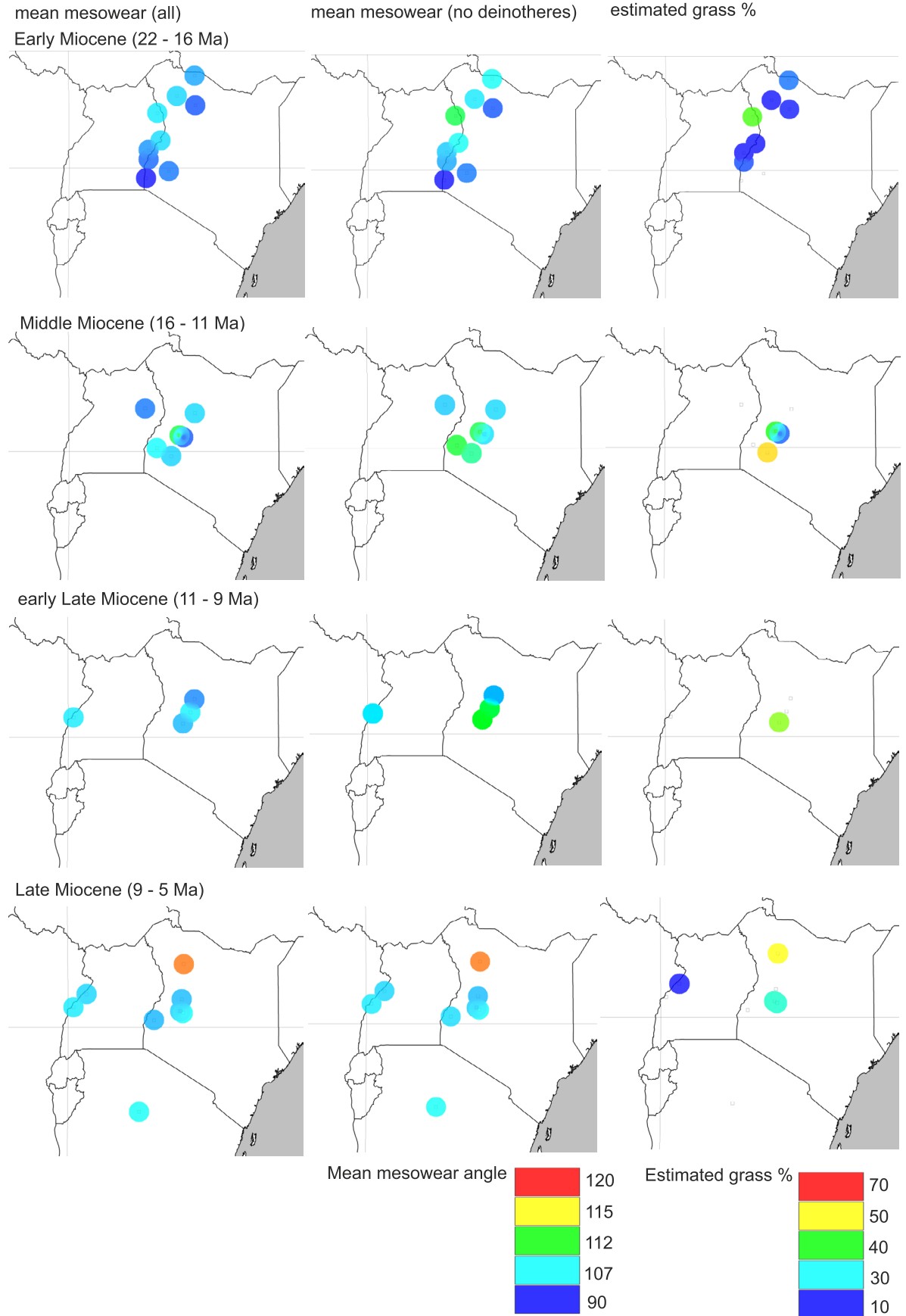

**Extended Data Fig. 4 | Colour-interpolated maps of mean mesowear angles of all proboscideans (left), elephantoids only (middle) and estimated grass cover (right) in East Africa during the Miocene.** A MapfInfo workbase file (world.base.wor) was used as the basemap.

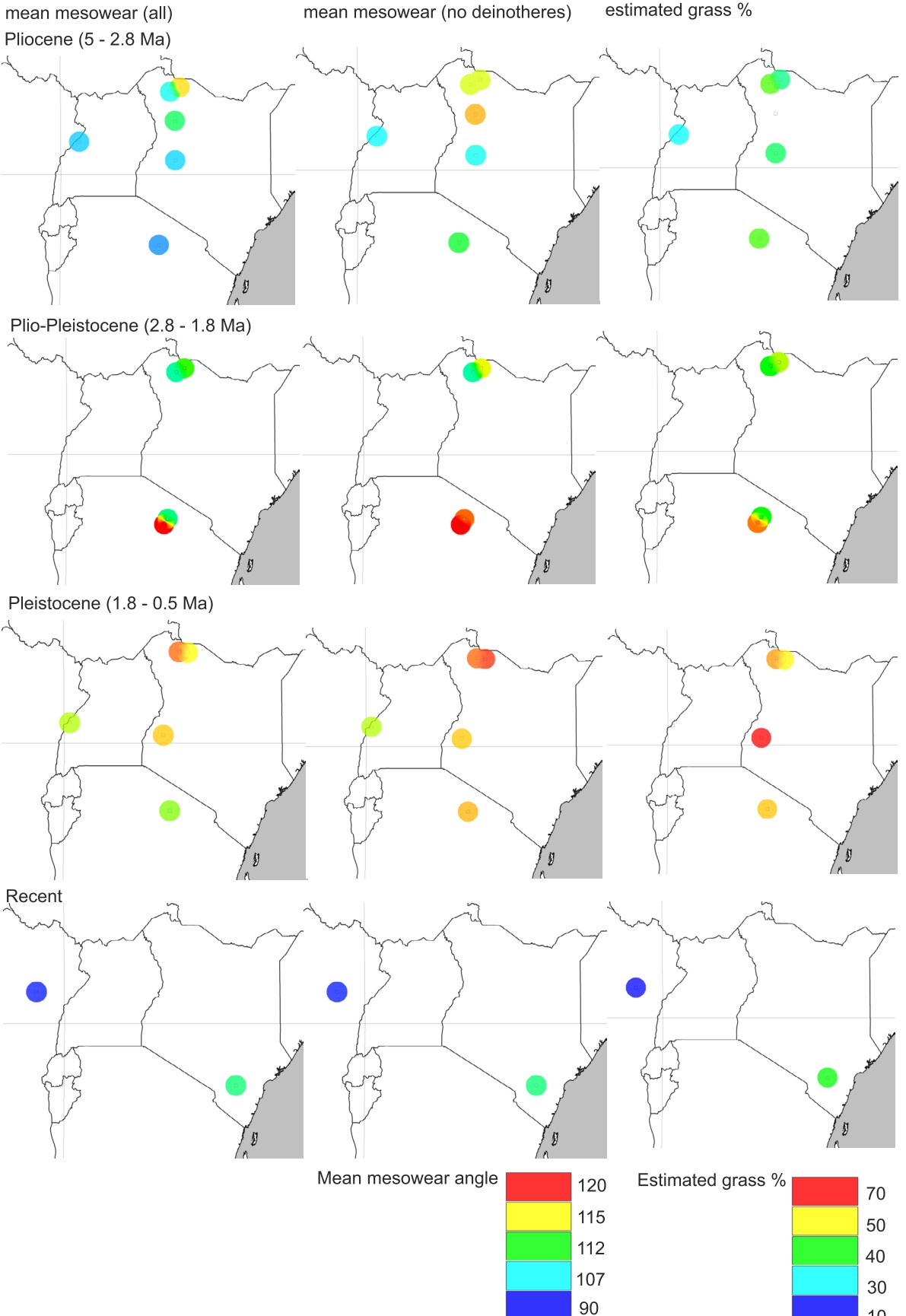

**Extended Data Fig. 5 | Colour-interpolated maps of mean mesowear angles of all proboscideans (left), elephantoids only (middle) and estimated grass cover (right) in East Africa from the Pliocene to the present.** A MapfInfo workbase file (world.base.wor) was used as the basemap.

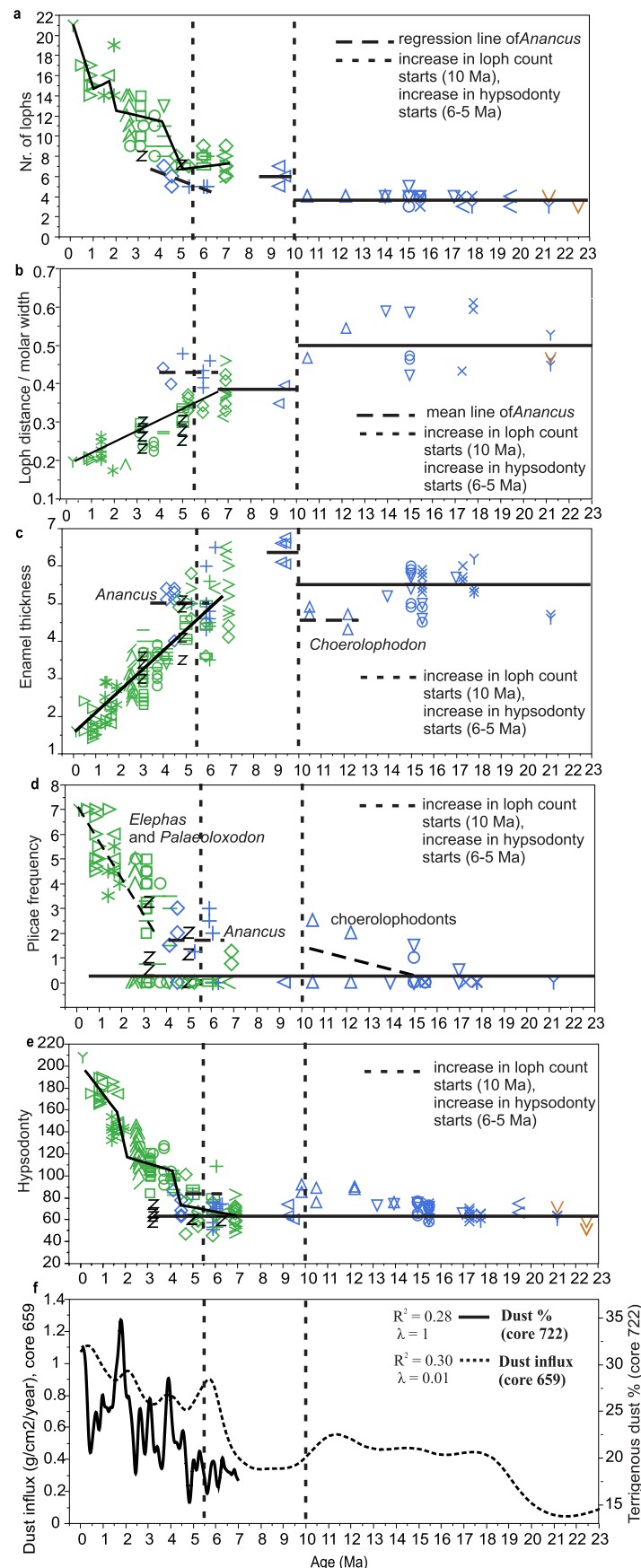

**Extended Data Fig. 6 | See next page for caption.**

**Extended Data Fig. 6 | The timing of dental trait shifts through time in relation to each another and aridity proxies.** Timing of morphological shifts across the Proboscidea (minus deinotheres) (**a-e**), and terrigenous dust accumulation (**f**) 23 - 0 Ma. **a**) number of lophs/lamellae, **b**) loph distance / molar width (inverse of LF*W; see Methods), **c**) enamel thickness, **d**) plicae frequency, **e**) hypsodonty and **f**) smoothing spline fit of core 659 dust influx data (dotted line) and core 722 dust % data[23]. Solid horizontal lines in (a)-(e) indicate means of 'gomphotheres' up to 10 or 5 Ma, and of *Tetralophodon* and/or early elephantids 10-5 Ma; solid diagonal lines indicate regression of main directional change since ca. 6-4 Ma in derived Elephantidae. Vertical dashed lines mark beginning of morphological shifts at 10 Ma (increased loph count and packing) and at 6-5 Ma (hypsodonty increase). Short dashed lines indicate means or regression lines for *Anancus* (**a** – **d**), *Loxodonta cookei* (**e**) and choerolophodonts (**d**).

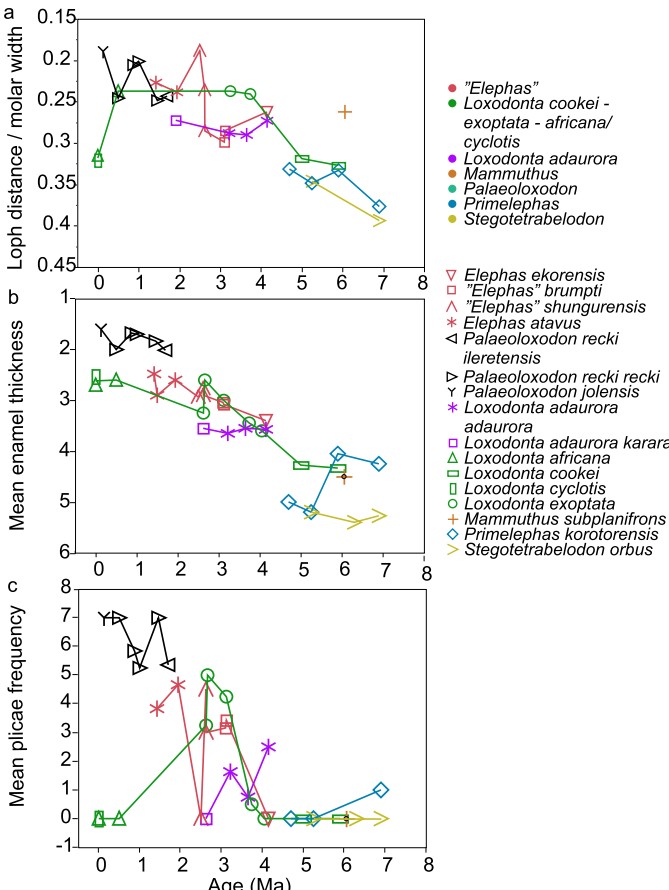

**Extended Data Fig. 7 | Trends across population means in loph distance / molar width (a), enamel thickness (b) and plicae frequency (c), separated by elephantid lineages 7 – 0 Ma.** Note that the y-axes for **(a)** and **(b)** are reversed showing decreasing values towards the top of the figure to aid visual comparison with the other traits. Compare with Fig. 4.

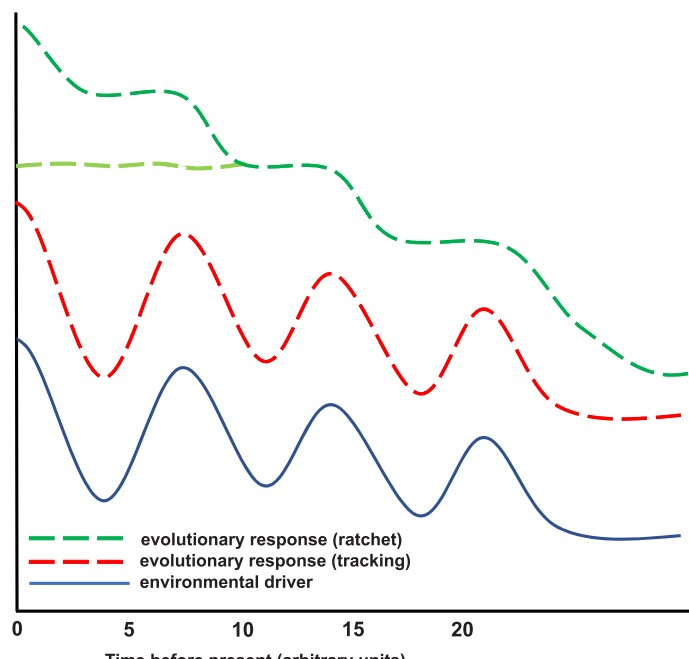

**Extended Data Fig. 8 | Schematic diagram showing possible evolutionary responses to an increasing but fluctuating environmental driver (selective force).** In the tracking response (red) evolutionary traits revert when the environment reverses. In the ratcheted response (green), the traits remain in stasis during environmental reversal. Pale green line: the trait has reached a maximal or optimal value after two increases and remains in stasis thereafter (as in *Loxodonta* in the present study).

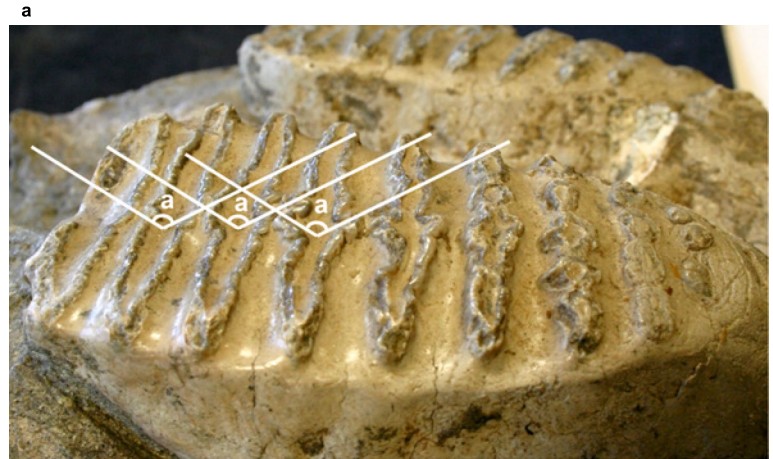

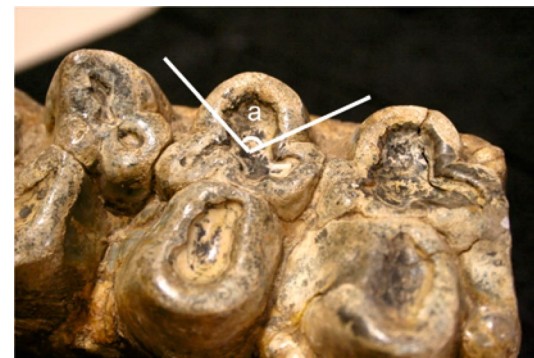

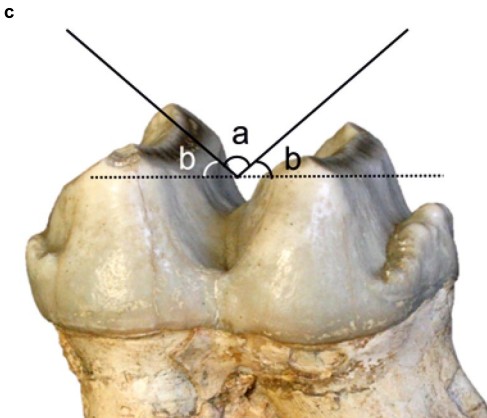

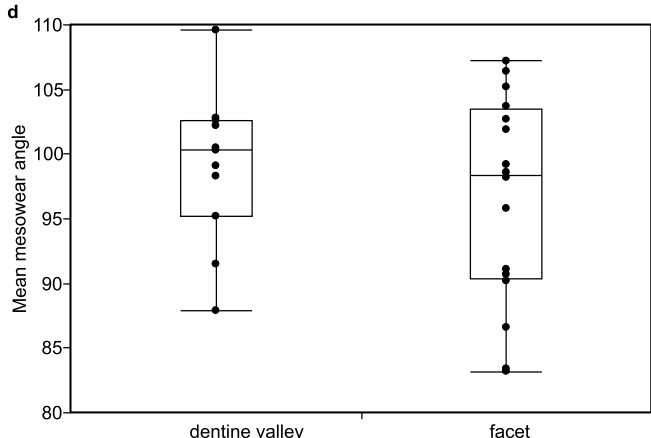

**Extended Data Fig. 9 | Measurement of mesowear angles (angle a) of proboscideans (a-c), and comparison of mesowear angles measured from dentine valleys vs. facet slopes in Deinotheriidae (d). a**) in Elephantidae the mesowear angles were measured on lamellae in medium wear stage by placing the tip of the angle at the bottom of a worn dentine valley and the sides of the angle tangent to the top of the adjacent enamel ridges[16]. Mean mesowear angle per tooth was calculated as a mean of angles measured from three central worn lamellae in the molar. **b**) In 'gomphotheres', the mesowear angles were measured similarly to elephantids, by placing the tip of the angle at the deepest point of worn dentine valley within a lophid, with the sides of the angle touching the tops of the enamel ridges. **c**) Mainly in Deinotheriidae (and some specimens of Mammutidae and 'Gomphotheriidae' s.l.), the mesowear angle data were complemented by angles measured from wear facets in worn lophs (angle b). The latter was measured by placing one side of the angle parallel to the level of the tooth crown base and the other side parallel to the slope of the facet, the

mesowear angle then calculated as a = 180°-2*b. Example specimens shown: **a**) M3 (dext.) of *Elephas atavus* (Elephantidae), Olduvai Bed I, Tanzania (NHMUK-PV-M14691), **b**) M3 (sin.) of *Protanancus macinnesi* (Amebelodontidae), Maboko, Kenya (NHMUK-PV-M15530) and **c**) m3 (sin.) of *Deinotherium bozasi*, Olduvai Bed II, Tanzania (NHMUK-PV-M14119). **d**) Two-sided pairwise Wilcoxon test comparison of mesowear angles measured from worn dentine valleys (n = 11) and from wear facets in enamel (n = 16) in Deinotheriidae. In the box plots, the central line within the box marks the median, the lower boundary of the box marks the first quartile (25th percentile), the upper boundary of the box marks the third quartile (75th percentile) and upper and lower lines of the whiskers mark the maximum and minimum values, respectively. The mean difference (2°) is non-significant (Z = −0.67, p = 0.5) and is within the margin of measurement error for an individual angle. Specimens photographed at the Natural History Museum, London.

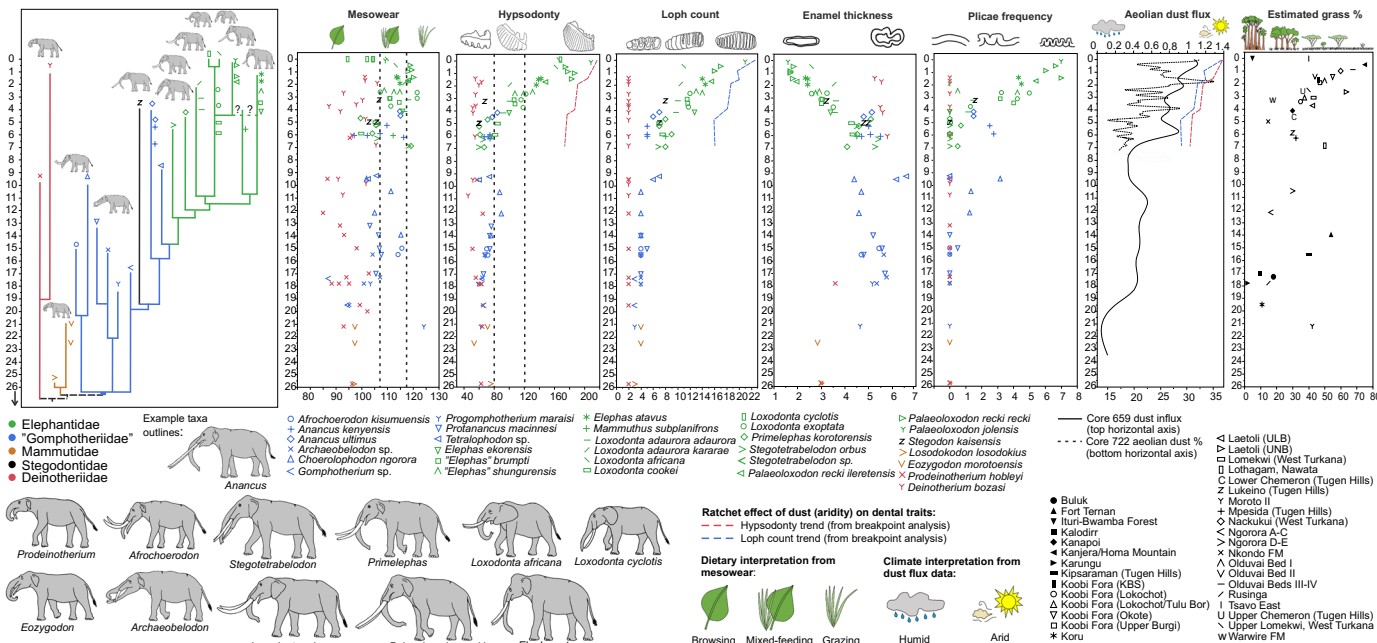

**Extended Data Fig. 10 | Summary figure.** Summary figure of phylogeny, diet and dental traits of Proboscidea, and environmental proxies, during the last 26 million years in East Africa. From left to right: East African proboscidean phylogeny based on Cantalapiedra et al.[58]; mean mesowear angles of proboscidean taxa; dental traits (hypsodonty, loph count, enamel thickness and mean plicae frequency [on 1 mm of enamel band]); aeolian dust records from deep-sea sedimentary cores; and grass % estimates of plant communities based on sources summarized in Supplementary Information and Supplementary Data. The vertical lines in the mesowear figure indicate approximate threshold values dividing browsing, mixed-feeding and grazing dietary signals. The vertical lines in the hypsodonty figure represent thresholds between brachydont, mesodont and hypsodont crown height categories. The patterns of average hypsodonty increase (red dashed line) and average loph count increase (blue line) during the last 7 Ma are shown to demonstrate the ratchet effect of the aridification peaks (core 722 dust data) on these dental traits. The outline reconstructions of example proboscidean taxa are made by J. Saarinen based on available craniodental and postcranial evidence from East African fossil record, in some cases completed based on additional information from closely related taxa.

# Reporting Summary

## Statistics

For all statistical analyses, confirm that the following items are present in the figure legend, table legend, main text, or Methods section.

| n/a | Confirmed | |
|---|---|---|
| ☐ | ☒ | The exact sample size ($n$) for each experimental group/condition, given as a discrete number and unit of measurement |
| ☐ | ☒ | A statement on whether measurements were taken from distinct samples or whether the same sample was measured repeatedly |
| ☐ | ☒ | The statistical test(s) used AND whether they are one- or two-sided *Only common tests should be described solely by name; describe more complex techniques in the Methods section.* |
| ☐ | ☒ | A description of all covariates tested |
| ☐ | ☒ | A description of any assumptions or corrections, such as tests of normality and adjustment for multiple comparisons |
| ☐ | ☒ | A full description of the statistical parameters including central tendency (e.g. means) or other basic estimates (e.g. regression coefficient) AND variation (e.g. standard deviation) or associated estimates of uncertainty (e.g. confidence intervals) |
| ☐ | ☒ | For null hypothesis testing, the test statistic (e.g. $F$, $t$, $r$) with confidence intervals, effect sizes, degrees of freedom and $P$ value noted *Give P values as exact values whenever suitable.* |
| ☒ | ☐ | For Bayesian analysis, information on the choice of priors and Markov chain Monte Carlo settings |
| ☒ | ☐ | For hierarchical and complex designs, identification of the appropriate level for tests and full reporting of outcomes |
| ☐ | ☒ | Estimates of effect sizes (e.g. Cohen's $d$, Pearson's $r$), indicating how they were calculated |

*Our web collection on statistics for biologists contains articles on many of the points above.*

## Software and code

Policy information about availability of computer code

| Data collection | NA |
|---|---|
| Data analysis | R Studio version 3.5.3, 4.2.2, JMP 14, IBM SPSS v. 25, STATISTICA 13.3 |

For manuscripts utilizing custom algorithms or software that are central to the research but not yet described in published literature, software must be made available to editors and reviewers. We strongly encourage code deposition in a community repository (e.g. GitHub). See the Nature Portfolio guidelines for submitting code & software for further information.

## Data

Policy information about availability of data

All manuscripts must include a data availability statement. This statement should provide the following information, where applicable:
- Accession codes, unique identifiers, or web links for publicly available datasets
- A description of any restrictions on data availability
- For clinical datasets or third party data, please ensure that the statement adheres to our policy

All data are provided in Supplementary Data Tables 1-10 and in Figshare online repository (10.6084/m9.figshare.23276126)

# Research involving human participants, their data, or biological material

Policy information about studies with human participants or human data. See also policy information about sex, gender (identity/presentation), and sexual orientation and race, ethnicity and racism.

| | |
|---|---|
| Reporting on sex and gender | NA |
| Reporting on race, ethnicity, or other socially relevant groupings | NA |
| Population characteristics | NA |
| Recruitment | NA |
| Ethics oversight | NA |

Note that full information on the approval of the study protocol must also be provided in the manuscript.

# Field-specific reporting

Please select the one below that is the best fit for your research. If you are not sure, read the appropriate sections before making your selection.

☐ Life sciences          ☐ Behavioural & social sciences          ☒ Ecological, evolutionary & environmental sciences

For a reference copy of the document with all sections, see nature.com/documents/nr-reporting-summary-flat.pdf

# Ecological, evolutionary & environmental sciences study design

All studies must disclose on these points even when the disclosure is negative.

| | |
|---|---|
| Study description | We analyse relationships between proboscidean dental traits, dental mesowear (diet) and environmental proxies (aeolian dust accumulation and locality mean ordinated hypsodonty for aridity, stable carbon isotopes and paleobotanical records for proportion of grass in plant communities), using least squares multiple regressions, multiple regressions commonality analyses, time series breakpoint analyses and partial correlations of dental trait variables. We explore phylogenetic correlations of traits with Pagel's test performed on global proboscidean supretree. |
| Research sample | The primary research material for this study are morphometric measurements and dental mesowear angles from East African fossil proboscidean molar teeth covering the last 26 million years. For associated data we use various literature sources for estimated percentage of grasses in fossil plant communities (based on paleobotanical and stable isotope records) and aeolian dust accumulation in deep-sea sediments (source literature listed in the supplementary material). In addition, we used NOW-database (https://nowdatabase.org/) for locality mean ordinated hypsodonty values (additional aridity proxy). For Pagel's test of phylogenetic correlation of traits we used proboscidean supertree data published by Cantalapiedra et al. (2021) |
| Sampling strategy | Availability of fossil material and associated environmental proxy data dictated sample sizes. Sample sizes were not restricted beyond the availability of fossil material and available proxy data. |
| Data collection | Juha Saarinen (corresponding author) collected the data during research visits to several palaeontological collections. J Saarinen also collected associated paleo-proxy data from literature and database sources. |
| Timing and spatial scale | The data were collected during museum visits between 2016 and 2019. |
| Data exclusions | No available data were excluded from the study. In cases where part of the data were excluded from particular analyses, reasons for excluding those data in those particular cases have been justified in the manuscript text. |
| Reproducibility | All analyses presented in the study can be reproduced using the data are provided in supplementary tables and the analysis methods, software and r-packages described in the manuscript. |
| Randomization | NA |
| Blinding | NA |

Did the study involve field work?          ☐ Yes          ☒ No

# Reporting for specific materials, systems and methods

We require information from authors about some types of materials, experimental systems and methods used in many studies. Here, indicate whether each material, system or method listed is relevant to your study. If you are not sure if a list item applies to your research, read the appropriate section before selecting a response.

## Materials & experimental systems

| n/a | Involved in the study |
|-----|----------------------|
| ☒ | Antibodies |
| ☒ | Eukaryotic cell lines |
| ☐ | ☒ Palaeontology and archaeology |
| ☒ | Animals and other organisms |
| ☒ | Clinical data |
| ☒ | Dual use research of concern |
| ☒ | Plants |

## Methods

| n/a | Involved in the study |
|-----|----------------------|
| ☒ | ChIP-seq |
| ☒ | Flow cytometry |
| ☒ | MRI-based neuroimaging |

## Palaeontology and Archaeology

**Specimen provenance**
The proboscidean fossils used in this study are stored in the following museums and institutes: Natural History Museum, London, UK; Humboldt Museum of Natural History, Berlin, Germany; Central Africa Museum, Tervuren, Belgium; National Museums of Kenya, Nairobi; Kipsaraman Museum, Kipsaraman,Kenya; Tsavo Research Station, Voi, Kenya; Uganda Museum, Kampala; and National Museum of Tanzania, Dar es Salaam). Research permits for studying the materials from Kenya and Tanzania were obtained from NACOSTI and COSTECH, respectively. JJS had research affiliations with the Kenya Wildlife Service, National Museums of Kenya, National Museum of Tanzania and Uganda Museum for the research.

**Specimen deposition**
All specimens are deposited in the collections of the museums listed above.

**Dating methods**
This study does not include new dating of the fossil materials. All data on fossil locality ages are based on published literature sources listed in the article and/or supplementary materials.

☒ Tick this box to confirm that the raw and calibrated dates are available in the paper or in Supplementary Information.

**Ethics oversight**
All museums and institutes storing the fossil materials used in this study agreed to the study protocol. Necessary research permits and affiliations were obtained from the museums and istitutions, and from NACOSTI for research done in Kenya and COSTECH for research done in Tanzania.

Note that full information on the approval of the study protocol must also be provided in the manuscript.

