## [Peer Review File · Nature Ecology & Evolution]

Peer Review Information

Journal: Nature Ecology & Evolution

Manuscript Title: Dietary innovation and fluctuating climate drove ratcheted evolution of proboscidean dental traits

Corresponding author name(s): Juha Saarinen

Editorial Notes:

Reviewer Comments & Decisions:

Decision Letter, initial version:

24th February 2023

Dear Dr Saarinen,

Your manuscript entitled "Adaptive trends and the evolutionary ratchet: decoupling the effects of diet and environmental change on proboscidean dental evolution" has now been seen by three reviewers, whose comments are attached. The reviewers have raised a number of concerns which will need to be addressed before we can offer publication in Nature Ecology & Evolution. We will therefore need to see your responses to the criticisms raised and to some editorial concerns, along with a revised manuscript, before we can reach a final decision regarding publication.

We note that reviewer 2 comments on the lack of causal evidence in the absence of phylogenetic tests--we invite you to consider adding suitable analyses to the manuscript as we think it might substantively strengthen it. Apart from that, and a concern from reviewer 1 that you need to more clearly discuss the role of grass phytoliths and dust in dental evolution, there are really only minor comments from the reviewers.

We therefore invite you to revise your manuscript taking into account all reviewer and editor comments. Please highlight all changes in the manuscript text file.

* If you have not done so already please begin to revise your manuscript so that it conforms to our Article format instructions at <http://www.nature.com/natecolevol/info/final-submission>. Refer also to any guidelines provided in this letter.

2* Include a revised version of any required reporting checklist. It will be available to referees (and, potentially, statisticians) to aid in their evaluation if the manuscript goes back for peer review. A revised checklist is essential for re-review of the paper.

[REDACTED]

Nature Ecology & Evolution is committed to improving transparency in authorship. As part of our efforts in this direction, we are now requesting that all authors identified as 'corresponding author' on published papers create and link their Open Researcher and Contributor Identifier (ORCID) with their account on the Manuscript Tracking System (MTS), prior to acceptance. ORCID helps the scientific community achieve unambiguous attribution of all scholarly contributions. You can create and link your ORCID from the home page of the MTS by clicking on 'Modify my Springer Nature account'. For more information please visit www.springernature.com/orcid.

[REDACTED]

Reviewer expertise:

Reviewer #1: eastern African palaeoenvironments and phytolith analysis

Reviewer #2: dental mesowear analysis

Reviewer #3: proboscidean evolution

Reviewers' comments:

Reviewer #1 (Remarks to the Author):

Dear authors,

I appreciate all the work and efforts you have put into this manuscript "Adaptive trends and the evolutionary ratchet: decoupling the effects of diet and environmental change on proboscidean dental evolution" which will contribute largely in our understanding of the role of habitats/diets in dental evolution of the elephants. However, it is not clear if this paper is a review article or a research report because key sections that would guide the reader are not clearly outlined. The flow of the information is not quite consistent and this needs to be addressed before the article is accepted for publication. Other minor revisions are suggested in the ms in the line-by-line review.

Best wishes,

Reviewer #2 (Remarks to the Author):

The MS reviews evidence of the record of African proboscidean dental morphology, wear dietary proxies and the relationship between these variables and proxies for regional and local climatic aridity and dust abundance. The data show clear relationships of certain variables (hypsodonty and others) with aridity and dust volume, while others such as enamel folding are more clearly correlated with diet variables (mesowear) and not aridity. These changes appear in a step-like fashion, suggesting the action of an evolutionary ratchet. The analysis is sound and elegant in showing a clear stepwise time series, and the arguments are convincing, but I would strongly suggest the authors back down a bit on the language suggesting key innovations or that this pattern is definitively causative because their analysis does not show either of these things - strong correlations, yes, suggestive, yes, but *not* evidence of causation without further testing. Neither they or the referred article by Saegusa have tested for key innovation using any phylogenetic tests. Further, this MS uses an awkward sentence referring to 'preadaptation', which is a problematic and teleological construction - I suggest the authors rethink the sentence as it can be interpreted badly. I have made suggestions in the text for some grammatical changes as well. These are all issues of framing language, not critiques of the methods or evidence, which seems to be solid and fascinating.

Reviewer #3 (Remarks to the Author):

This is an ambitious work tracing patterns in the evolution of proboscidean dentition, with some unexpected results, e.g. grazing in early elephantoids such as *Progomphotherium* - which I take at face value, mesowear analysis being outside the scope of my expertise.

I have the following comments:

- There is some inconsistency in the taxonomy used. The first paragraph of the Supplementary Information (lines 8-10) lists the elephantoid families Mammutidae, Stegodontidae, Gomphotheriidae, Amebelodontidae, Tetralophodontidae, Anancidae and Elephantidae (inexplicably omitting Choerolophodontidae); at the same time, Fig. 1 in the main text has only Elephantidae, Gomphotheriidae, Mammutidae and Stegodontidae, and so do tables S.1-S4 in the Excel file.

3Obviously, "Gomphotheriidae" in the figure and tables corresponds to 'gomphotheres' as used throughout the text, i.e. a paraphyletic assemblage and should probably be replaced with the latter to avoid confusion. Protanancus (l. 180-181) belongs to Amebelodontidae and not Gomphotheriidae (again, I understand the implied meaning of "gomphothere". Similarly, (l. 82) " the gomphothere Choerolophodon" should probably become "the 'gomphothere' Choerolophodon".

- l. 149: obviously third molars are meant but this is only mentioned on l. 612.

- l. 247/8: was increase in loph connected with aridification in Stegodontidae too?

- l. 269: I am not quite sure if this relates to Elephantidae only or to elephantoids in general? Choerolophodon has thick but plicated enamel.

- l. 346: Since there are no known true elephants at 9 Ma I suppose this relates to lines 147-149, "The derived "gomphothere" Tetralophodon is the likely sister-group of Elephantidae and close to its ancestry, and samples dated to 10-9 Ma are taken as the starting-point for subsequent morphological evolution"? Perhaps a clarification, or a reminder, on l. 346 would be helpful for the reader.

- Supplementary information file:

- l. 13: The idea of Tetralophodon as the stem group of elephants can be traced, I believe, at least to the mid-80s, in a works by Pascal Tassy.

- l. 45, "loph-like structures called lophids": I am not sure what is meant here. Lophs are structures of the upper teeth, and lophids of the lower ones.

- l. 58: lamellae narrower – obviously, antero-posteriorly shorter is meant but can be potentially confusing in my opinion.

- l. 60: for late Miocene Stegodon in Africa see e.g. Sanders et al. (2010)

None of these are vital to the manuscript's contents but addressing them might make it more easily accessible for the reader. I would recommend a minor revision.

Georgi N. Markov

Author Rebuttal to Initial comments

Responses to reviewers (NB line numbers in our responses are those of the revised ms).

Reviewer 1:

"Dear authors,

I appreciate all the work and efforts you have put into this manuscript "Adaptive trends and the evolutionary ratchet: decoupling the effects of diet and environmental change on proboscidean dental evolution" which will contribute largely in our understanding of the role of habitats/diets in dental evolution of the elephants. However, it is not clear if this paper is a review article or a research report because key sections that would guide the reader are not clearly outlined. The flow of the information is not quite consistent and this needs to be addressed before the article is accepted for publication. Other minor revisions are suggested in the ms in the line-by-line review."

Response: We thank the reviewer for these suggestions. Concerning the structure of the article: because the two other reviewers have not commented on this and because of the lack of more detailed information on where the structure is hard to follow, we leave the decision on this matter to the Editor. We are prepared to modify the structure of the text if needed, though we would prefer not to make major structural changes at this point. This is a research article where we present a lot of new data and analyses, so it is certainly not intended to be a review article. We feel this is self-evident, but to emphasise the point we have made explicit in the Abstract and in the Introduction that we present new data (lines 15 & 49).

"Adaptive trends and the evolutionary ratchet: decoupling the effects of diet and environmental change on proboscidean dental evolution is an important contribution to our understanding of the role of habitats and fluctuating environments in the proboscidean dental evolution in East Africa over the last 26 Ma. However, the ms has major hiccup in the structure and flow of the paper. It is difficult for a reader to capture the approach and methodologies presented in the paper."

Response: See previous response. Regarding the approach and methodologies, the detailed Methods are performed at the end of the text and in the supplements, but we have given an easy-to-understand summary of the approach and methodology at the start of each section (e.g. lines 60-70, 146-149, 207-235, 296-298).

"References are a bit mix up, for instance one would expect the referencing to start with 1 moving forward but this is not the case."

Response: We thank the reviewer for noting this. The reason for this is that the references start in the abstract (references 1 and 2 are first mentioned there), so the reference numbering in the beginning of Introduction continues from there. In case this seems confusing, we have added a short passage at the beginning of Introduction (lines 34-39) to introduce references 1 and 2 there as well. This is also a helpful way of starting the introduction, as it introduces the key concepts and questions relevant to the study.

"In addition, the role of grass phytoliths and dust in dental evolution is not clearly demonstrated."

Response: We have discussed the role of phytoliths and dust as factors that increase dental wear rates, acknowledging that both factors have played a role in the evolution of the dental traits associated with increased aridity (see lines 42-44, 207-217). In our statistical analyses, the strong association of dental traits with dust flux implicates aridity as a major driver, but we did not set out to resolve whether external mineral dust or phytoliths were more important for dental evolution, because it has been shown in several other studies that both have an effect on tooth wear, and both are prominent factors in seasonally dry environments. Instead, we explore the effects of aridity itself, in comparison with dietary changes, on the evolution of dental traits. Nonetheless our results do suggest dust as the dominant proximate factor, while

not discounting the contribution of phytoliths even though their relative importance is difficult to quantify, and this is made clear at lines 274-279.

"I suggest the authors to carefully review the ms and present it in way the reader can understand without going back and forth between paragraphs. Make use of short sentences. Both spatial and temporal comparison of the fossils studied has not been done well. Also, it is not clear if this is a review paper or a research paper. If the above and the following minor line-by-line revisions are addressed, the paper can be resubmitted for review."

Response: We thank the reviewer for these suggestions. We have conformed with all the requests of the referee where specific text was indicated (see below). We have also split some long sentences (e.g. lines 344-348). The issue of review versus primary research has also been clarified (see above). Beyond this, again as above, we would need more concrete suggestions if more substantial modifications to the structure of the text are required.

Line 35: *Why do the referencing start at 3 instead of 1?*

Response: See above.

Line 63: *This is not clear*

Response: We assume the reviewer refers to the description of the "lithogenic flux"? We have now clarified this in the text (line 68).

Line 69-71: *"This indicated predominantly browsing diet in an array of mammalian orders including proboscideans, with a shift from browsing to grazing with the C3-C4 transition at around 10-8 Ma in East Africa14,21" This needs re-wording for clarity*

Response: We thank the reviewer for noting this and we have now clarified the sentence (line 74).

Lines 84-85: *"Grass pollen is not a strong distinguishing proxy between grass beyond family level. How did you identify the abundance of C3 grasses and sure this is not C4 grasses as well?"*

Response: The evidence behind this conclusion can be found in the research articles we cite in the text. The grasses at Fort Ternan have been identified as mostly C3 grasses based on stable carbon isotope studies (see discussion in Retallack et al., 1992). In fact, the pollen analysis indicates the presence of grass families that today are C4 photosynthesizing at Fort Ternan, but the stable isotope analyses do not show evidence of any major component of C4 plants in Fort Ternan, even if there now is evidence of the presence of C4 grasses already from the Early Miocene (Peppe et al. 2023).

Lines 89-90: *"Here you need to elaborate that these grass habitats resemble modern grass glades (open grass patches that are found in forests)"*

Response: Many thanks for this suggestion, we have modified the text to clarify this (lines 98-99).

Line 100: *"true elephants" is this an acceptable term to describe the modern elephants? Are there false elephants- just curious.*

Response: By this we mean members of the family Elephantidae (as opposed to other elephantoids), and we have now added this to the sentence for clarity (line 107).

Lines 105-109: *This sentences need revision so that the reader can understand your intended message.*

Response: The sentence at lines 105-7 read "This elicited an increase in the grass component of the diet (Figs. 1, 2a) with potential selective pressure on dental morphology". We are uncertain which part of this

source, provide a link to the Creative Commons license, and indicate if changes were made. In the cases where the authors are anonymous, such as is the case for the reports of anonymous peer reviewers, author attribution should be to 'Anonymous Referee' followed by a clear attribution to the source work. The images or other third party material in this file are included in the article's Creative Commons license, unless indicated otherwise in a credit line to the material. If material is not included in the article's Creative Commons license and your intended use is not permitted by statutory regulation or exceeds the permitted use, you will need to obtain permission directly from the copyright holder. To view a copy of this license, visit <http://creativecommons.org/licenses/by/4.0/>.

was unclear, but in case the word 'elicited' was the problem we have changed it to 'led to' for simplicity (now line 115). Line 108 was blank and 109 began the caption of Fig. 1, – these are not associated with the sentence on lines 105-107.

Fig. 2: *do you mean grass-dominated and mixed feeding or is this meant to be one category*

Response: We assume the reviewer refers to the horizontal dashed line in Fig 2a. Indeed, it marks the threshold between browsing (below the line) and mixed-feeding to grazing (above the line) diets. We have revised the figure caption to clarify this (lines 134-136).

Line 137: *For consistency purposes, use one style for your dates/age. Either Myr or MA*

Response: We thank the reviewer for the suggestion. However, we think there is a slight conceptual difference in the usage of Ma vs. Myr. The abbreviation Ma is usually used to refer to age ("millions of years old"), whereas Myr is a more generalized abbreviation of "millions of years". We thus believe that "Myr" is the correct abbreviation in the context of this sentence. For clarity, we have added an explanation of these usages in the Methods (lines 687-688).

Line 142-143: *Not sure I understand*

Response: Here we simply say that expanding the diet into more mixed-feeding in a grassy habitat is associated with elevated hypsodonty and thinner, more plicated enamel in *Choerolophodon* compared with other gomphotheres (thus paralleling such associations in elephants). We have modified the sentence to clarify this (lines 151-154).

Figures: *be consistent with fonts and styles for all the figures*

Response: Many thanks for the remark. We have updated the fonts for all the figures according to the formatting instructions of Nature Ecology & Evolution.

Line 325: *Refer to the previous comments*

Response: Again, it is hard to know what is required here, but perhaps the sentence spanning lines 324-8 was overly long. We have split it to simplify comprehension (new lines 345-348).

Lines 330-332: *Restructure the sentence, "Hence speciation facilitates directional evolution "by retaining, stepwise, the advances made in any one direction. Successive speciation events are the pitons attached to the slopes of an adaptive peak"49"*

Response: This sentence (new lines 348-350) contains (in quotation marks) a direct quotation from Futuyma's article (new ref 50), and thus restructuring it would lose the point of directly citing his prosaic metaphor.

Reviewer 2:

"The MS reviews evidence of the record of African proboscidean dental morphology, wear dietary proxies and the relationship between these variables and proxies for regional and local climatic aridity and dust abundance. The data show clear relationships of certain variables (hypsodonty and others) with aridity and dust volume, while others such as enamel folding are more clearly correlated with diet variables (mesowear) and not aridity. These changes appear in a steplike fashion, suggesting the action of an evolutionary ratchet.

*The analysis is sound and elegant in showing a clear stepwise time series, and the arguments are convincing, but I would strongly suggest the authors back down a bit on the language suggesting key innovations or that this pattern is definitively causative because their analysis does not show either of these things - strong correlations, yes, suggestive, yes, but *not* evidence of causation without further testing."*

"Neither they or the referred article by Saegusa have tested for key innovation using any phylogenetic tests".

- **comment on row 18:** *" This is a strong statement. You have a clear tight correlation pattern, and a very plausible mechanism, but can we be absolutely certain that it led to these changes?"*
- **comment on row 19:** *"Again, these are really hard to convincingly test - has anyone done so for this? Saegusa's results are suggestive but he applied no statistical or phylogenetic tests to support his contention of a key innovation"*

Response: We thank the reviewer for these positive and constructive remarks, and fully agree on the issue of causality. Our analyses showed correlation between diet and dental traits, and causal relationships were argued based on the order in which these changes happened (e.g., the 'pro-al' chewing mechanism preceded changes in dental traits, and peaks in aridity were coincident with or slightly preceded successive increases in loph count and hypsodonty). Nonetheless, we agree that causality is a complex issue, and to answer this concern we have now applied Pagel's Test of phylogenetically correlated evolution of traits (Pagel, 1994) to test Saegusa's hypothesis of proal chewing as a key adaptation permissive of the evolution of other dental traits. In this we used the proboscidean supertree published by Cantalapiedra et al. (2021), where corresponding author J. Saarinen was also a coauthor (lines 676-682 and new Supplementary Information 4). Our results (Suppl. 4 & lines 166-169) support a model where increase in loph count (to 6 or more) in derived proboscideans was indeed dependent on the evolution of proal chewing. This model had higher likelihood than ones where these traits were independent, depended on each other, or proal chewing depended on loph count.

This result supports the idea that the increase in loph count was indeed causally related to major changes in the masticatory system into a proal chewing cycle, so the latter was, as others have supposed, a "key innovation" allowing improvement in molar shearing efficiency to deal with vegetational changes brought by aridification. Conversely, the results do not provide support for the evolution of hypsodonty being also consequent on pro-al chewing, at least not directly; it was rather a later adaptation to increase functional durability of the molars. We have taken care not to go beyond these findings (lines 166-169, 378-394, 411-415), and have obliterated the use of the word 'key' that the reviewer considered 'loaded' (see below).

Our second conclusion on causality concerned the correlation between later trends in the dental traits and dust flux (aridity proxy). Although the reviewer only touched on this, we here show how we have addressed it.

1) We had already quantitatively tested the temporal relationship using breakpoint analysis (Fig. 4), that showed clear temporal correlation between peaks in dust flux and increments in dental traits. We have investigated further methods that compare time-series of dependent and independent variables, but it would be difficult to apply them to our data because the time series of dental traits have unavoidable gaps resulting from the nature of the fossil record. In any case, by the very nature of the ratchet pattern we uncovered, these methods would fail because the curves are not expected to match perfectly; only *increases* in the independent variable drive changes in the dependent one; this is why we employed breakpoint analysis.

As far as inferring causality from this correlation, there are three possibilities: (i) aridity drove dental evolution, (ii) dental evolution drove aridity, (iii) aridity and dental evolution were driven by a third factor. Option (ii) can be discounted. Option (iii) is possible but we do not impute the dust proxy as the causal factor, but aridity more generally - a deliberately high-level environmental factor - and it is hard to envisage what at a higher level still (e.g. tectonics?) could have independently impacted dental evolution of terrestrial mammals as well as triggering arid climate. We therefore argue that the pattern of stepwise increase in hypsodonty and loph count that clearly correlates to phases of increased dust flux suggests a causal relationship from some aspect of aridification to the evolution of these dental traits (option i). We have taken care with our wording, e.g.:

- Line 238 & 257-259: dental traits are said only to 'have significant relationships with' 'correlate to' or 'relate to' aridity proxies
- Line 263-4 & 274 "our results... *suggest* aridity as the main driver of proboscidean hypsodonty"
- Line 313-14: we state that the breakpoint results "strongly corroborate aridity as a driver of dental evolution" and we stand by that, but could alter the wording if required. We are basically positing a reasoned hypothesis, open to be supported or falsified in the future.

2) Our data for diet and dental traits of proboscideans includes a spatial pattern in addition to a temporal one. Our combination of time series and locality-based analyses support the hypothesis of causal relationship of aridification and diet on dental evolution, as we first see major changes in dental traits in localities that indicate locally more arid conditions than in other contemporaneous parts of East Africa (for example, in the Turkana region in Late Miocene), and these also tend to be areas with the first evidence of expansion of grass-rich plant communities and grazing diets. See lines 107-111, Supplementary Information 3.

We have explained this logic in a new section, 'A note on causality' in Supplementary Information 4.

"I would strongly suggest the authors back down a bit on the language suggesting key innovations"

- **comment on row 137:** *"What makes them key? Kind of a loaded word in the context"*

Response: We have modified or removed our use of this term in all instances. For example:

- Line 18 (Abstract): We wrote: "We show that behavioural experimentation in diet correlated to environmental context, and that it led to major adaptive change in dental traits but only after acquisition of a key innovation in the masticatory system". This refers only to the fact that early experiments in grazing didn't lead to much dental modification, but once pro-al chewing had been invented (in elephant precursors) it did. Still to avoid the k-word we replaced 'a key innovation' by 'functional innovations'.
- Line 146 (former 137): Here 'key' was being used in its literal sense: "We mapped the evolution of key dental adaptations in proboscidean molars over 26 Myr". But given its more particular meaning in 'key innovation' we changed it from 'key' to 'major'.
- Line 165: here we are simply citing (without affirming) Saegusa's stated hypothesis that pro-al is a 'key adaptation'. We made this clearer by adding '*that has been considered* the key adaptation...'

source, provide a link to the Creative Commons license, and indicate if changes were made. In the cases where the authors are anonymous, such as is the case for the reports of anonymous peer reviewers, author attribution should be to 'Anonymous Referee' followed by a clear attribution to the source work. The images or other third party material in this file are included in the article's Creative Commons license, unless indicated otherwise in a credit line to the material. If material is not included in the article's Creative Commons license and your intended use is not permitted by statutory regulation or exceeds the permitted use, you will need to obtain permission directly from the copyright holder. To view a copy of this license, visit <http://creativecommons.org/licenses/by/4.0/>.

- Line 168: in the summary of the new Pagel test results, we avoided 'key' and said 'likely depended on'
- Line 381 we have replaced 'key innovations' by just 'innovations'
- Line 681 'key adaptation' replaced by 'evolution'

"Further, this MS uses an awkward sentence referring to 'preadaptation', which is a problematic and teleological construction - I suggest the authors rethink the sentence as it can be interpreted badly. I have made suggestions in the text for some grammatical changes as well. These are all issues of framing language, not critiques of the methods or evidence, which seems to be solid and fascinating."

- **comment on rows 160-163:** *"I think pre-adaptation is a really poor word choice here (see Gould and Vrba, 1982, Exaptation—a missing term in the science of form). It suggests foreknowledge and is misleading. Instead, it would be better to say that the acquisition of proal chewing allowed elephants to begin exploitation of grasses, allowing them to shift into grassier habitats and further changes driven by aridity. Using preadaptation gets the potential causation backwards."*

Response: We thank the reviewer for this insightful comment, and we agree. We have now reworded the sentence, replacing 'predapted' by 'enabled' (line 174).

Reviewer 3:

"This is an ambitious work tracing patterns in the evolution of proboscidean dentition, with someone unexpected results, e.g grazing in early elephantoids such as Progomphotherium – which I take at face value, mesowear analysis being outside the scope of my expertise."

I have the following comments:

- There is some inconsistency in the taxonomy used. The first paragraph of the Supplementary Information (lines 8-10) lists the elephantoid families Mammutidae, Stegodontidae, Gomphotheriidae, Amebelodontidae, Tetralophodontidae, Anancidae and Elephantidae (inexplicably omitting Choerolophodontidae); at the same time, Fig. 1 in the main text has only Elephantidae, Gomphotheriidae, Mammutidae and Stegodontidae, and so do tables S.1-S4 in the Excel file. Obviously, "Gomphotheriidae" in the figure and tables corresponds to 'gomphotheres' as used throughout the text, i.e. a paraphyletic assemblage and should probably be replaced with the latter to avoid confusion. Protanancus (l. 180-181) belongs to Amebelodontidae and not Gomphotheriidae (again, I understand the implied meaning of "gomphotheres". Similarly, (l. 82) "the gomphotheres Choerolophodon" should probably become "the 'gomphotheres' Choerolophodon"."

Response: We thank the reviewer for noting this, and indeed those remarks are all correct. We have used "Gomphotheriidae" both in the broad sense to refer to the paraphyletic "gomphotheres" (in the figures) and in the stricter sense (in the supplement). In the phylogenetic tree we use blue colour to mark the lineages called "gomphotheres" in the broad sense, as in the figures. To clarify this matter and avoid misunderstandings, we have changed the word "Gomphotheriidae" in the figures into "Gomphotheriidae s.l.", and listed the families that this paraphyletic group includes in the caption of Fig. 1. We also thank the reviewer for noting the lack of Choerolophodontidae in the supplementary materials, we have now added that.

source, provide a link to the Creative Commons license, and indicate if changes were made. In the cases where the authors are anonymous, such as is the case for the reports of anonymous peer reviewers, author attribution should be to 'Anonymous Referee' followed by a clear attribution to the source work. The images or other third party material in this file are included in the article's Creative Commons license, unless indicated otherwise in a credit line to the material. If material is not included in the article's Creative Commons license and your intended use is not permitted by statutory regulation or exceeds the permitted use, you will need to obtain permission directly from the copyright holder. To view a copy of this license, visit <http://creativecommons.org/licenses/by/4.0/>.

- l. 149: obviously third molars are meant but this is only mentioned on l. 612.

Response: Many thanks for noting this, we have added it (line 160).

- l. 247/8: was increase in loph connected with aridification in Stegodontidae too?

Response: This is a very good question, but in the scope of the present study we can't fully answer this, because the early evolution of the stegodonts happened outside Africa, in Eastern and Southern Asia. The dispersal of stegodonts into Africa does broadly correspond with the Late Miocene onset of aridification there, but on the other hand the environmental evidence associated with *Stegodon kaisensis* suggest that it occupied relatively wooded environments (and its diet was clearly browse-dominated). It is possible that stegodonts could have originated in relatively open and dry environments in Asia, but the scarcity of early stegodont fossil record currently prevents this to be extensively explored. Saegusa (2020) noted that the earliest stegodonts with fully developed proal chewing and increased loph count are associated with relatively wooded paleoenvironments and he leaves this question open, but he notes that aridification (or perhaps rather increased seasonality?) is a possible factor behind the origination of the derived traits in stegodonts.

- l. 269: I am not quite sure if this relates to Elephantidae only or to elephantoids in general? *Choerolophodon* has thick but plicated enamel.

Response: This (inverse correlation of plicae to enamel thickness) does refer to Elephantidae only, based on the elephantid data in Suppl Tables 6 & 7 (mentioned in new line 285). We have clarified this in line 288. Indeed, *Choerolophodon* had thicker enamel than most elephantids, but compared to other gomphotheres, it does have somewhat reduced enamel thickness.

- l. 346: Since there are no known true elephants at 9 Ma I suppose this relates to lines 147-149, "The derived "gomphotheres" *Tetralophodon* is the likely sister-group of Elephantidae and close to its ancestry, and samples dated to 10-9 Ma are taken as the starting-point for subsequent morphological evolution"? Perhaps a clarification, or a reminder, on l. 346 would be helpful for the reader.

Response: Indeed, that's correct, we refer to *Tetralophodon* as the sister group and possible ancestor of Elephantidae. However, the origination of Elephantidae is a bit ambiguous due to scarcity of earliest material, and in fact Saegusa et al. (2014) considered a partial molar from Nakali (ca. 9.8 Ma) to represent the earliest known true elephantid. However, the dietary shift does indeed happen somewhere between *Tetralophodon* (browser) and earliest Lothagam elephantids (grazers) between about 9 and 7 Ma. We have added a reminder of this here (lines 366-9).

- Supplementary information file:

- l. 13: The idea of *Tetralophodon* as the stem group of elephants can be traced, I believe, at least to the mid-80s, in a works by Pascal Tassy.

Response: Thanks, that's correct, we have added some references here (line 16).

- l. 45, “loph-like structures called lophids”: I am not sure what is meant here. Lophs are structures of the upper teeth, and lophids of the lower ones.

Response: We thank the reviewer for noting this. Here we meant both upper and lower loph-like structures. We have modified this sentence to be more accurate in this sense (lines 44-45).

- l. 58: lamellae narrower – obviously, antero-posteriorly shorter is meant but can be potentially confusing in my opinion.

Response: Many thanks, we have now added the word “antero-posterior” to specify that we mean narrowing in that direction (line 58).

- l. 60: for late Miocene *Stegodon* in Africa see e.g. Sanders et al. (2010)

Response: Many thanks for noting this. That’s correct, the earliest *Stegodon* in Africa is from the Late Miocene. We have now corrected this (line 61).

None of these are vital to the manuscript’s contents but addressing them might make it more easily accessible for the reader. I would recommend a minor revision.

Decision Letter, first revision:

19th May 2023

Dear Dr. Saarinen,

Thank you for submitting your revised manuscript "Adaptive trends and the evolutionary ratchet: decoupling the effects of diet and environmental change on proboscidean evolution" (NATECOLEVOL-23010129A). It has now been seen again by the original reviewers and their comments are below. The reviewers find that the paper has improved in revision, and therefore we'll be happy in principle to publish it in Nature Ecology & Evolution, pending minor revisions to satisfy the reviewers' final requests and to comply with our editorial and formatting guidelines.

[REDACTED]

Reviewer #1 (Remarks to the Author):

The revised version of the MS "Adaptive trends and the evolutionary ratchet: decoupling the effects of diet and environmental change on proboscidean dental evolution" has been greatly improved and clearly present the relationship of dental traits with aridity and dust. The sentence structure improves the clarity of the significance of the research for both the expert and non-expert in the subject. I strongly think that the manuscript is ready for publication.

Reviewer #2 (Remarks to the Author):

The authors have done an outstanding job - they went beyond my request to consider that causality had not been demonstrated by going ahead and testing it. The changes in language and the careful discussion of both the analysis and the difficulties of inferring causation were very well handled and have allayed my concerns. I look forward to publication.

13Reviewer #3 (Remarks to the Author):

I thank the authors for their responses and accept them.
I think publication is now justified, yet, and excuse the nitpicking:

While the new version has added "lophs (in upper teeth) or lophids (in lower teeth), from now on collectively referred to as "lophs" " (Suppl. Information I. 45), elsewhere lophs and lophods are still used somewhat inconsistently (e.g. I. 62 in the main text, or 611, 647 etc. in Methods); Tassy (1986) is listed in the main references (I. 814) but I couldn't find it mentioned in the text; I still think gomphothere (e.g. I. 88, referring to Choerolophodon) should be in quotation marks, and the authors might decide if they should be double (e.g. I. 158) or single (e.g. I. 417).

Georgi Markov

Our ref: NATECOLEVOL-23010129A

26th May 2023

Dear Dr. Saarinen,

Thank you for your patience as we've prepared the guidelines for final submission of your Nature Ecology & Evolution manuscript, "Adaptive trends and the evolutionary ratchet: decoupling the effects of diet and environmental change on proboscidean evolution" (NATECOLEVOL-23010129A). Please carefully follow the step-by-step instructions provided in the attached file, and add a response in each row of the table to indicate the changes that you have made. Please also check and comment on any additional marked-up edits we have proposed within the text. Ensuring that each point is addressed will help to ensure that your revised manuscript can be swiftly handed over to our production team.

****We would like to start working on your revised paper, with all of the requested files and forms, as soon as possible (preferably within two weeks). Please get in contact with us immediately if you anticipate it taking more than two weeks to submit these revised files.****

When you upload your final materials, please include a point-by-point response to any remaining

14reviewer comments.

In recognition of the time and expertise our reviewers provide to Nature Ecology & Evolution's editorial process, we would like to formally acknowledge their contribution to the external peer review of your manuscript entitled "Adaptive trends and the evolutionary ratchet: decoupling the effects of diet and environmental change on proboscidean evolution". For those reviewers who give their assent, we will be publishing their names alongside the published article.

Nature Ecology & Evolution offers a Transparent Peer Review option for new original research manuscripts submitted after December 1st, 2019. As part of this initiative, we encourage our authors to support increased transparency into the peer review process by agreeing to have the reviewer comments, author rebuttal letters, and editorial decision letters published as a Supplementary item. When you submit your final files please clearly state in your cover letter whether or not you would like to participate in this initiative. Please note that failure to state your preference will result in delays in accepting your manuscript for publication.

Cover suggestions

As you prepare your final files we encourage you to consider whether you have any images or illustrations that may be appropriate for use on the cover of Nature Ecology & Evolution.

Nature Ecology & Evolution has now transitioned to a unified Rights Collection system which will allow our Author Services team to quickly and easily collect the rights and permissions required to publish your work. Approximately 10 days after your paper is formally accepted, you will receive an email in providing you with a link to complete the grant of rights. If your paper is eligible for Open Access, our Author Services team will also be in touch regarding any additional information that may be required

15to arrange payment for your article.

Please note that *Nature Ecology & Evolution* is a Transformative Journal (TJ). Authors may publish their research with us through the traditional subscription access route or make their paper immediately open access through payment of an article-processing charge (APC). Authors will not be required to make a final decision about access to their article until it has been accepted. [Find out more about Transformative Journals](https://www.springernature.com/gp/open-research/transformative-journals)

Authors may need to take specific actions to achieve [compliance with funder and institutional open access mandates](https://www.springernature.com/gp/open-research/funding/policy-compliance-faqs). If your research is supported by a funder that requires immediate open access (e.g. according to [Plan S principles](https://www.springernature.com/gp/open-research/plan-s-compliance)) then you should select the gold OA route, and we will direct you to the compliant route where possible. For authors selecting the subscription publication route, the journal's standard licensing terms will need to be accepted, including [self-archiving-and-license-to-publish](https://www.nature.com/nature-portfolio/editorial-policies/self-archiving-and-license-to-publish). Those licensing terms will supersede any other terms that the author or any third party may assert apply to any version of the manuscript.

[REDACTED]

[REDACTED]

Reviewer #1:

Remarks to the Author:

The revised version of the MS "Adaptive trends and the evolutionary ratchet: decoupling the effects of diet and environmental change on proboscidean dental evolution" has been greatly improved and clearly present the relationship of dental traits with aridity and dust. The sentence structure improves the clarity of the significance of the research for both the expert and non-expert in the subject.

16I strongly think that the manuscript is ready for publication.

Reviewer #2:

Remarks to the Author:

The authors have done an outstanding job - they went beyond my request to consider that causality had not been demonstrated by going ahead and testing it. The changes in language and the careful discussion of both the analysis and the difficulties of inferring causation were very well handled and have allayed my concerns.

I look forward to publication.

Reviewer #3:

Remarks to the Author:

I thank the authors for their responses and accept them.

I think publication is now justified, yet, and excuse the nitpicking:

While the new version has added "lophs (in upper teeth) or lophids (in lower teeth), from now on collectively referred to as "lophs" " (Suppl. Information l. 45), elsewhere lophs and lophods are still used somewhat inconsistently (e.g. l. 62 in the main text, or 611, 647 etc. in Methods); Tassy (1986) is listed in the main references (l. 814) but I couldn't find it mentioned in the text; I still think gomphothere (e.g. l. 88, referring to Choerolophodon) should be in quotation marks, and the authors might decide if they should be double (e.g. l. 158) or single (e.g. l. 417).

Georgi Markov

Final Decision Letter:

5th July 2023

Dear Dr Saarinen,

We are pleased to inform you that your Article entitled "Fluctuating climate and dietary innovation drove ratcheted evolution of proboscidean dental traits", has now been accepted for publication in Nature Ecology & Evolution.

Over the next few weeks, your paper will be copyedited to ensure that it conforms to Nature Ecology and Evolution style. Once your paper is typeset, you will receive an email with a link to choose the appropriate publishing options for your paper and our Author Services team will be in touch regarding any additional information that may be required

17After the grant of rights is completed, you will receive a link to your electronic proof via email with a request to make any corrections within 48 hours. If, when you receive your proof, you cannot meet this deadline, please inform us at rjsproduction@springernature.com immediately.

You will not receive your proofs until the publishing agreement has been received through our system

Due to the importance of these deadlines, we ask you please us know now whether you will be difficult to contact over the next month. If this is the case, we ask you provide us with the contact information (email, phone and fax) of someone who will be able to check the proofs on your behalf, and who will be available to address any last-minute problems . Once your paper has been scheduled for online publication, the Nature press office will be in touch to confirm the details.

Acceptance of your manuscript is conditional on all authors' agreement with our publication policies (see www.nature.com/authors/policies/index.html). In particular your manuscript must not be published elsewhere and there must be no announcement of the work to any media outlet until the publication date (the day on which it is uploaded onto our web site).

Please note that *Nature Ecology & Evolution* is a Transformative Journal (TJ). Authors may publish their research with us through the traditional subscription access route or make their paper immediately open access through payment of an article-processing charge (APC). Authors will not be required to make a final decision about access to their article until it has been accepted. [Find out more about Transformative Journals](https://www.springernature.com/gp/open-research/transformative-journals)

Authors may need to take specific actions to achieve [compliance](https://www.springernature.com/gp/open-research/funding/policy-compliance-faqs) with funder and institutional open access mandates. If your research is supported by a funder that requires immediate open access (e.g. according to [Plan S principles](https://www.springernature.com/gp/open-research/plan-s-compliance)) then you should select the gold OA route, and we will direct you to the compliant route where possible. For authors selecting the subscription publication route, the journal's standard licensing terms will need to be accepted, including [those licensing terms](https://www.nature.com/nature-portfolio/editorial-policies/self-archiving-and-license-to-publish) will supersede any other terms that the author or any third party may assert apply to any version of the manuscript.

An online order form for reprints of your paper is available at http://www.springernature.com/reprints

<https://www.nature.com/reprints/author-reprints.html>><https://www.nature.com/reprints/author-reprints.html>. All co-authors, authors' institutions and authors' funding agencies can order reprints using the form appropriate to their geographical region.

We welcome the submission of potential cover material (including a short caption of around 40 words) related to your manuscript; suggestions should be sent to Nature Ecology & Evolution as electronic files (the image should be 300 dpi at 210 x 297 mm in either TIFF or JPEG format). Please note that such pictures should be selected more for their aesthetic appeal than for their scientific content, and that colour images work better than black and white or grayscale images. Please do not try to design a cover with the Nature Ecology & Evolution logo etc., and please do not submit composites of images related to your work. I am sure you will understand that we cannot make any promise as to whether any of your suggestions might be selected for the cover of the journal.

You can generate the link yourself when you receive your article DOI by entering it here: <http://authors.springernature.com/share>.

[REDACTED]

P.S. Click on the following link if you would like to recommend Nature Ecology & Evolution to your librarian <http://www.nature.com/subscriptions/recommend.html#forms>

** Visit the Springer Nature Editorial and Publishing website at http://editorial-jobs.springernature.com?utm_source=ejp_NEcoE_email&utm_medium=ejp_NEcoE_email&utm_campaign=ejp_NEcoE>www.springernature.com/editorial-and-publishing-jobs for more information about our career opportunities. If you have any questions please click [here](mailto:editorial.publishing.jobs@springernature.com).**